# In-situ plasmonic tracking oxygen evolution reveals multistage oxygen diffusion and accumulating inhibition

Jun-Gang Wang[1], Lifang Shi[2], Yingying Su[1,3], Liwei Liu[2], Zhenzhong Yang[4], Rong Huang[4], Jing Xie[5], Yang Tian[1] & Di Li [1,3✉]

Understanding mass transfer processes concomitant with electrochemical conversion for gas evolution reactions at the electrode-electrolyte interface plays a key role in advancing renewable energy storage and conversion. However, due to the complicated diffusion behavior of gas at the dynamic catalytic interfaces, it is still a great challenge to accurately portray mass transfer of gas during electrocatalysis process. Here, we track the diffusion of dissolved oxygen on Cu nanostructured plasmonic interface, which reveals multistage oxygen diffusion behaviors, including premature oxygen accumulation, spontaneous diffusion and accelerated oxygen dissipation. This work uncovers an accumulating inhibition effect on oxygen evolution arising from interfacial dissolved oxygen. With these knowledges, we develop a programmable potential scan strategy to eliminate interfacial gas products, which alleviates the concentration polarization, releases accessible actives sites and promotes electrocatalytic performance. Our findings provide a direct observation of the interfacial mass transfer processes that governs the kinetics of gas-involved multiphases catalysis.

[1] School of Chemistry and Molecular Engineering, East China Normal University, Shanghai 200241, China. [2] Institute of Optics and Electronics, Chinese Academy of Sciences, Chengdu 610209, China. [3] Key Laboratory of Bioorganic Phosphorus Chemistry and Chemical Biology, Department of Chemistry, Tsinghua University, Beijing 10084, China. [4] Key Laboratory of Polar Materials and Devices (MOE), Department of Electronics, East China Normal University, Shanghai 200241, China. [5] Institute of Microelectronics, Chinese Academy of Sciences, Beijing 100029, China. ✉email: dli@chem.ecnu.edu.cn

Electrode–electrolyte interfaces (EEIs), where mass transfer and energy transfer proceeds concomitantly, are of paramount significance in electrochemistry[1–3]. Oxygen evolution reaction (OER) plays a pivotal role in water splitting by providing protons and electrons at the EEIs; however, its sluggish kinetics limit their performance and commercialization[4–6]. Gas evolution leads to an undesirable blockage of redox reaction sites and ion conducting pathways, resulting in an increase of ohmic resistance and formation of a heterogeneous concentration gradient at the EEIs[7,8]. All these effects give rise to energy losses and unfavorable attenuation of electrochemical conversion efficiencies. Moreover, under different oxygen concentration gradients, the electrode interfaces experience different oxygen compositions ($OH^-$, $OH^*$, dissolved oxygen, oxygen bubble, etc.) at various polarization. This, in turn, generates varying operation regimes in terms of either mass transport or kinetic reaction limitations for the electrocatalytic processes, leading to confusing interfacial reaction mechanisms[9]. The dynamics of oxygen concentration gradients at the EEIs thus remain quite elusive, as it is readily hidden by vast spectators and electrolytes, and is often overlooked during kinetic analysis. Therefore, it is of crucial importance to precisely identify various oxygen diffusion stages under operando conditions and uncover the influence of oxygen concentration gradients on OER performance[10].

Conventional electrochemical measurements, however, are insufficient to identify dissolved oxygen and uncover the dynamic dissolved oxygen diffusion process, because of the short of direct molecular recognition ability and the complexity induced by non-Faradaic processes, such as concomitant mass transfer of spectator species and multi-phase transition. In situ spectro-electrochemical techniques[11–13], such as X-ray absorption/diffraction/scattering spectroscopy[14,15], infrared and Raman spectroscopy[16], mass spectrometry[17,18], and nuclear magnetic resonance[19], are powerful tools for identifying accessible catalytic active sites, adsorbed intermediates, and monitoring the reconstruction of electrocatalysts under operando conditions. Unfortunately, these approaches also lack the ability of identifying fluctuations of interfacial dissolved oxygen due to the inherent negligible weak interaction between dissolved oxygen and the electrode surface. Thus, it is still a great challenge for current techniques to in situ observe oxygen evolution on heterogeneous catalytic interfaces and distinguish the diffusion profile of dissolved oxygen within the interfacial electric field.

Plasmonic-enhanced spectroscopy[20–22] exploits the excellent optical performance of rationally designed plasmonic nanostructures and possesses noninvasive feature with superior sensitivity and compatibility in monitoring a variety of chemical processes, including heterogeneous catalysis[23–25], electrochemistry[26–28], and phase transition[29–31]. Thereby, plasmonic-enhanced spectroscopy may possess the ability to differentiate oxygen compositions during OER at EEIs. In the present study, we establish a dual-functional plasmonic Cu nanoparticle (NP) interface that is amenable to host electrochemical reaction and synchronous plasmonic extinction dissection. Harnessing the environment refractive index-sensitive plasmonic nanostructures, in situ extinction spectroscopy enables identification of oxygen composition changes during OER by analyzing small spectral variations induced by the dynamics of interfacial dissolved oxygen profile at the EEIs[32,33]. By comparing synchronized plasmonic and electrochemical responses, we uncover not only a multistage of oxygen diffusion at the EEIs, but also an accumulating inhibition effect from confined mass transfer of oxygen on OER. The identification of multistage of oxygen diffusion and accumulating inhibition effect suggests that removing the freshly evolved $O_2$ from active sites may accelerate the charge-transfer process and recover catalytic sites, leading to improved OER performance. We thereby develop a programmable potential scan strategy to electrochemically reduce freshly generated $O_2$, to prevent its accumulation at the EEIs. The proposed programmable potential scan strategy results in significantly improved OER performance, which shows a potential to be a general operation in OER experiments. With the ability to deliver critical interfacial reaction information, the proposed strategy will provide in-depth information for understanding the intrinsic potential-dependent dynamic processes of various gas-involving systems at the EEIs.

## Results

**Seed-mediated electrodeposition of plasmonic Cu NPs.** Conventional electrochemical measurements are incapable of in situ probing the oxygen diffusion behavior because of (i) severe concentration polarization induced by the increased chemical potential of dissolved gas molecules at the electrode interface and (ii) perturbations from non-Faradaic portion and concomitant side interfacial reactions at the electrode surface–liquid/gas diffusion layer, thereby unable to reveal the oxygen evolution dynamics at the EEIs. By taking advantage of the near-field enhancement, plasmonic-enhanced extinction spectroscopy provides an alternative way for real-time monitoring OER and revealing oxygen profiles under polarization.

Here we developed a plasmonic Cu NP electrode as a platform to explore OER. This platform was served as both an electrocatalytic unit and spectroscopically addressable component to host multiple in situ spectroelectrochemical techniques. Briefly, Cu NPs were grown on conductive indium tin oxide (ITO) substrates through a seed-mediated electrodeposition (Fig. 1a; for detailed discussions, see Supplementary Figs. 1–5). Atomic force microscopy (AFM) images of Cu NPs at different deposition cycles (Fig. 1b) indicated that Cu nuclei (ca. 34 nm in diameter) appeared and they were homogeneously distributed on ITO after the initial seeding process (Fig. 1b, i). An instantaneous nucleation was suggested as the dominating process instead of progressive nucleation[34]. With the increase of electrodeposition cycles, possible fusions of Cu nuclei in the proximity occurred, resulting in an enlargement of particle size; Cu NPs grew from 34 nm seeds at 0 cycle to maximum 234 nm at 450 cycles (Fig. 1b, c and Supplementary Fig. 6; for a detailed composition analysis, see Supplementary Fig. 7). In addition, a particle-stacked film was formed because of high Cu nuclei coverage, as confirmed by the high-resolution AFM image (Fig. 1d).

We then examined the extinction spectra of Cu NP electrodes as a function of electrodeposition cycles to reveal the correlation between Cu NP structure and its optical response (Fig. 1e). Once a conductive film interface is formed beneath Cu NPs, its plasmonic wave function involves a superposition of a charge-transfer plasmon (CTP) mode in monopolar modes of individual NPs. As the separation distance between adjacent NPs and junction resistance reduces, CTP experiences a remarkable monotonic redshift[35,36]. Therefore, the noticeable redshift of plasmonic resonance band in Fig. 1e was assigned to the conductive connection between Cu NPs and the underlying film. Meanwhile, another plasmonic resonance mode at around 450 nm remained stable, whereas its intensity increased with the electrodeposition cycles, which was attributed to the Ohmic dissipation associated with the charge transport across the junction and the reduction of the capacitive coupling[37].

Furthermore, finite-difference time-domain simulations provide more details about the plasmonic response of the NP-on-conductive film stacked structure. A strong coupling between the adjacent Cu NPs excited by a plane wave and polarized toward the interparticle axis gave rise to a low-energy CTP resonance band at 690 nm, promoted by the conduction through the substrate, and a high-energy screened-bonding plasmonic

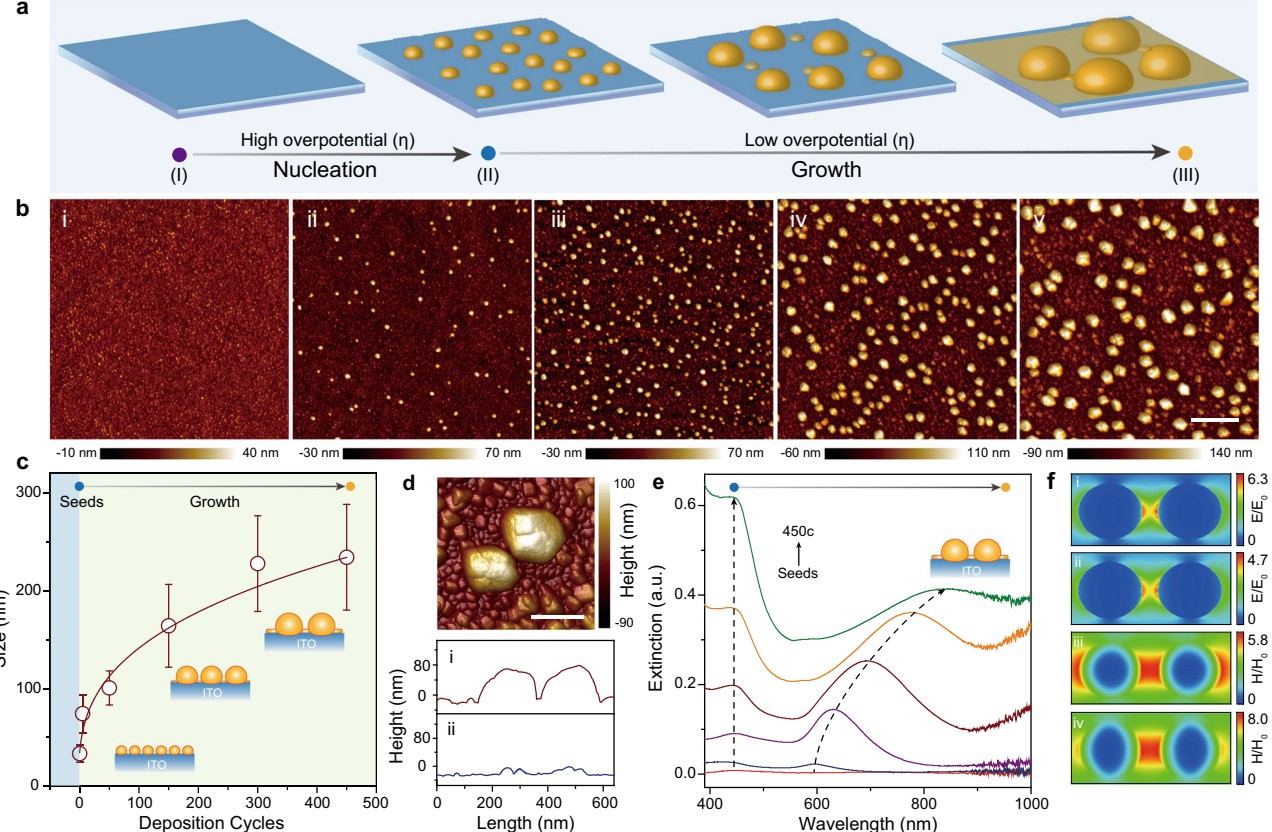

**Fig. 1 Seed-mediated electrodeposition of plasmonic Cu nanoparticles. a** Schematic illustrating the electrochemical seed-mediated growth strategy. **b** AFM images of Cu seeds (i) and Cu NPs at different deposition cycles (ii, 5; iii, 50; iv, 150; and v, 300 cycles) (scale bar: 1 μm). **c** The correlation between Cu NPs size and deposition cycles. Error bar represents the SE of Cu NPs size. **d** High-resolution AFM image of Cu NPs (top, 150 cycles, scale bar: 200 nm) and linear scan across Cu NPs dimmer (i) and nearby regions (ii) (bottom). **e** Ex situ extinction spectra of Cu seeds and Cu NPs at different deposition cycles (5, 50, 150, 300, and 450 cycles). **f** Top view of electric (i and ii) and magnetic (iii and iv) field snapshots for the near-field confinement at the plasmonic resonance wavelengths (i, iii, 450 nm; ii, iv, 690 nm).

resonance band at 450 nm (Fig. 1e, f). For both plasmonic modes, the induced electric and magnetic fields were confined in the vicinity of the gap region[38,39]. The experimentally recorded extinction spectra revealed a good agreement with the simulated extinction spectra (Supplementary Figs. 8 and 9). A more complicated model considering an isolated trimer or a more complicated chain structure could make the simulation results closer or more compatible to the real optical performance of the ensemble system; the possible multi-configurations of these structure result in extreme difficulty to precisely distinguish their contribution to ensemble optical performance at the present stage (Supplementary Fig. 10). The energy of CTP mode is known to be sensitive to the change of environmental refractive index; therefore, it is essential to convert subtle environmental change in gas evolution reaction into dynamic changes of extinction spectra.

**Dissolved oxygen diffusion profile monitored by a single-wavelength plasmonic strategy.** The beneficial interplay between large electrochemical surface area and interfacial high active species contributed to the high catalysis performance of copper materials (for detailed electrochemical measurements and discussion of oxygen evolution, see Supplementary Figs. 11 and 12). The plasmonic Cu NP electrode was then used to optically exploit oxygen diffusion behavior at the EEIs by taking advantage of the refractive index-sensitive plasmonic structure (Fig. 2a). Electrochemical steady-state extinction spectroscopy was first employed

to monitor OER (Fig. 2b and Supplementary Fig. 13). Under enhanced anodic polarization from −1.10 to −0.30 V (vs. Pt, all the potentials were referenced to a Pt quasi-reference electrode (QRE) unless otherwise stated), we observed a remarkable increase of extinction peak intensity (ca. 15%) and a 27 nm redshift of extinction peak (from 623 to 650 nm; Supplementary Fig. 13), attributing to the electrochemically formed Cu oxides and hydroxides (Supplementary Fig. 11). Notably, the extinction intensity at 623 nm increased ca. 14.9% at −0.30 V compared to that at −1.10 V. The extinction peak intensity decreased by ca. 70.5% and shifted to 720 nm from −0.30 to 0.90 V (Supplementary Fig. 13). These indicate that the Cu oxide and hydroxide intermediates were weakly bound and were readily dissolved during long-time anodic polarization, which hampered the interfacial process dissection under steady-state conditions.

We then developed a single-wavelength plasmonic monitoring strategy to precisely identify the diffusion behavior of dissolved oxygen (Fig. 2c). In general, the interfacial processes at the EEIs could induce a fluctuation of nearby refractive index, leading to a systematic transition from an original state ($S_1$, $I_1$–$\lambda_1$–$E_1$) to a reaction state ($S_2$, $I_2$–$\lambda_2$–$E_2$). Therefore, the fluctuation of the extinction intensity at a constant $\lambda$ during dynamic potential scan provides an ultrasensitive feedback of the interfacial reaction information at the EEIs (for detailed discussions of this concept, see Supporting Information Section 5). Of note, the NP-on-conductive film structure benefits the super sensitivity of the CTP mode ($\lambda_{CTP}$), which enables a precise recognition of delicate interfacial processes at the EEIs. Comparison of temporal

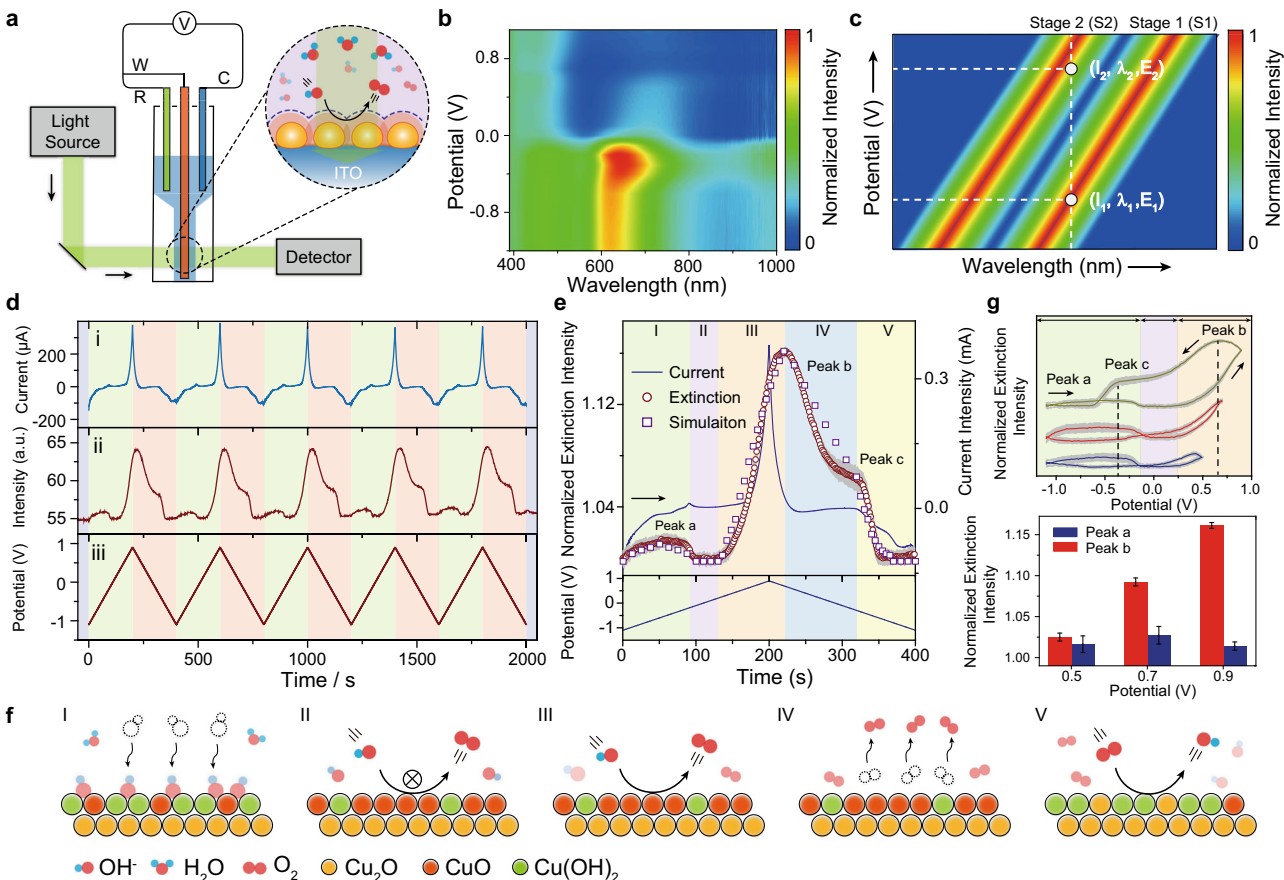

**Fig. 2 Plasmonic monitoring oxygen evolution reaction. a** Experimental setup of in situ electrochemical extinction microscopy. **b** The normalized dynamic extinction intensity (contour plots) of a Cu NPs electrode under steady-state anodic polarization. **c** Schematic illustration of the plasmonic single-wavelength strategy for real-time monitoring OER at the EEIs. **d** The consecutive electrochemical current (i) during OER and synchronous extinction signal (ii) under dynamic potential scan (iii). **e** The representative dynamic extinction trajectory (o), current curve (−), and FDTD simulation (□) for OER on a plasmonic Cu NPs electrode scanned from −1.10 to 0.90 V at a scan rate 0.01 V s$^{-1}$. **f** Schematic illustration of the interfacial reaction mechanism during dynamic potential scan. Note that both copper oxides and hydroxides serve as the real reactive species to react with OH$^−$ and then generate oxygen. **g** The extinction voltammetry with different terminational potential (0.50, 0.70, 0.90 V) at a scan rate of 0.01 V s$^{-1}$ (top panel). The dark region represents the standard error during consecutive five scan. The potential dependent of extinction intensity for peak *a* and *b* (bottom panel). Error bar represents the SE during consecutive five scans.

extinction trajectories at various wavelength during consecutive potential scan (Supplementary Fig. 14), the intensity increased ca. 1.60 and 2.33 times at 690 nm (CTP) compared to that of 443 nm and 560 nm, respectively. Therefore, high-sensitivity tracking of the electrochemical interfacial reaction at 690 nm (CTP) enables to explicitly identify characteristic spectral regions for revealing the OER mechanism. Series control experiments with Cu NP electrode performed under various potential scan range and scan rate revealed negligible extinction fluctuations (Supplementary Figs. 15–22). Despite the fact that weak anodic polarization current peak assigned to oxygen evolution was identified on bare ITO electrodes, the temporal optical readout cannot be readily observed in the absence of plasmonic-enhancement elements.

Accordingly, cyclic voltammetry and in situ extinction voltammetry measurements were performed synchronously on a home-made three-electrode micro-electrochemical cell, to explore oxygen evolution profile during OER. The reversible fluctuation of extinction intensity was in good agreement with current response (Fig. 2d). These periodic behaviors were attributed to the dynamic structural changes of the EEIs that involves both electrochemical redox of Cu NPs and oxygen evolution. Comparing this temporal extinction trajectory with the current response, five typical potential regions were identified to

explain the dynamic interfacial processes (Fig. 2e and Supplementary Fig. 11). These regions were as follows: (I) active region, where the oxidation of Cu occurs; (II) passive region, where oxygen evolution is suppressed; (III) transpassive region, where the onset oxygen evolution region is controlled by reactive kinetics; (IV) diffusion region, where oxygen diffusion regions is dominated by diffusion kinetics; and (V) reductive region, where oxygen reduction occurs (Fig. 2e, f). In particular, in Region I, although the potential was scanned from −1.10 to −0.36 V, the extinction intensity gradually increased under enhanced anodic polarization, indicating the phase growth associated within the oxide layer, ascribed to a temporal evolution of Cu$_2$O and CuO/Cu(OH)$_2$ phases (detailed X-ray photoelectron spectroscopy (XPS) analysis shown in Supplementary Fig. 23). In contrast, a rapid attenuation of the extinction intensity was observed after −0.36 V. These observations implied that potential-driven composition evolution of Cu oxides and hydroxides provide abundant vacancy sites for surface adsorption of OH to facilitate oxygen evolution through diminished OH/O ratio by releasing H to form desirable crystallization, which was further confirmed by quasi in situ time-of-flight secondary ion mass spectroscopy (ToF-SIMS) measurements (Supplementary Figs. 24 and 25)[40,41]. During the passive regions (> −0.10 V), indistinctive extinction

fluctuation indicated that the ratio of oxide to hydroxide species remained stable in the oxide layer (II).

Interestingly, when the potential was swept to transpassive regions (>0.20 V, III), the transformation from an amorphous oxide of CuO/Cu(OH)$_2$ phases to a crystalline structure generated sufficient electron-conductive paths through the anodic oxide film for oxygen evolution. Therefore, the rapid increase in extinction intensity could be reasonably attributed to the oxygen evolution[42]. Notably, when comparing extinction voltammetry with the linear potential scan from 0 to 0.90 V, a significantly lower onset potential for oxygen evolution was observed in extinction measurements (ca. 0.40 V). To understand the origin of this distinction in onset potential, both current and extinction voltammetry were fitted based on a one-dimensional diffusion model (see derivation in Supplementary Figs. 26 and 27).

$$\eta_c = \frac{RT}{nF} ln\left(\Phi(OH^-)\left(1-\frac{i}{i_{l,OH^-}}\right)\right)^4 \left(\Phi(O_2)\left(1-\frac{i}{i_{l,O_2}}\right)\right)^{-1}$$

$$(1)$$

where $T$ is the temperature, $R$ is the gas constant, $n$ is the electron transfer number, and $F$ is the Faraday's constant, $i_{l,\,OH^-}$ and $i_{l,\,O2}$ are the limiting current density of OH$^-$ oxidation (anodic polarization) and O$_2$ reduction (cathodic polarization), respectively. The partition coefficient $\Phi$ indicates the solubility of a species in the diffusion layer and can be defined as the ratio of the equilibrium concentration in the diffusion layer to the value in the bulk electrolyte[43].

From the fitting line, the concentration overpotential incurred by the oxygen diffusion layer was estimated to be ca. 0.20 V. Under enhanced oxygen concentration polarization, the measured current signals comprised not only the Faradaic components but also the non-Faradaic portion, both contributing to the remarkable distinction in the onset potential. As the extinction intensity is directly correlated with the amounts of interfacial dissolved oxygen, it represents only Faradaic current that diminishes interference from non-Faradaic contribution; therefore, this enables a precise identification of intrinsic onset potential at an early stage of the OER. Subsequently, the

extinction intensity showed a gradual attenuation with decreasing of potential (0.67 to −0.63 V), indicating the diffusion of dissolved oxygen far away from the EEIs. It was worth noting that the diffusion of dissolved oxygen followed two different modes (Fig. 2e, IV and V), a spontaneous convection with a decay rate of $1.0 \times 10^{-3}$ s$^{-1}$ (0.67 to −0.35 V), and an accelerated diffusion with a higher decay rate of $2.5 \times 10^{-2}$ s$^{-1}$ (−0.35 to −0.63 V). Due to the enhanced cathodic polarization, the newly formed interfacial species (Cu$_2$O/Cu(OH)$_2$) and near-negative electric field diminishes the desorption of oxygen and accelerates the dissipation of dissolved oxygen near the electrode surface.

We then established an interfacial reaction model through simulations of adsorbed chemical species and potential-dependent gas evolution at the EEIs to illustrate how the dynamic interfacial composition affects the plasmonic extinction spectroscopy (Fig. 2e). As the enhancement of anodic polarization, the phase transition from Cu$_2$O to CuO and Cu(OH)$_2$ phases led to the reduction of the refractive index from 2.90 to 2.72 and 1.71, resulting in an increase in the extinction intensity[44,45]. Further deprotonation in CuO/Cu(OH)$_2$ phases caused the growth of refractive index from 1.71 to 2.72, which accounts for the gradual attenuation in extinction intensity. When the oxygen evolution occurred, the dissolved oxygen accumulated at the electrode surface leads to a decrease of the local electrolyte refractive index, evidenced by the promptly increased spectroscopic intensity (Supplementary Fig. 28). More importantly, the spontaneous convection and accelerated diffusion of oxygen on the electrode surface during cathodic polarization resulted in a recovery of local interfacial environment. The corresponding simulated spectroscopic trends were in good accordance with experimental results, indicating that the proposed interfacial reaction model captured the intrinsic features of the OER on the plasmonic electrode interface. These results explicitly illustrated the dynamic oxidation of Cu plasmonic surface and uncovered the presence of multistage diffusion of dissolved oxygen at the EEIs. In addition, the extinction peak $b$ that appeared at 0.67 V was assigned to the oxygen evolution at the EEIs and the intensity of this peak increased quickly with enhancement of anodic polarization

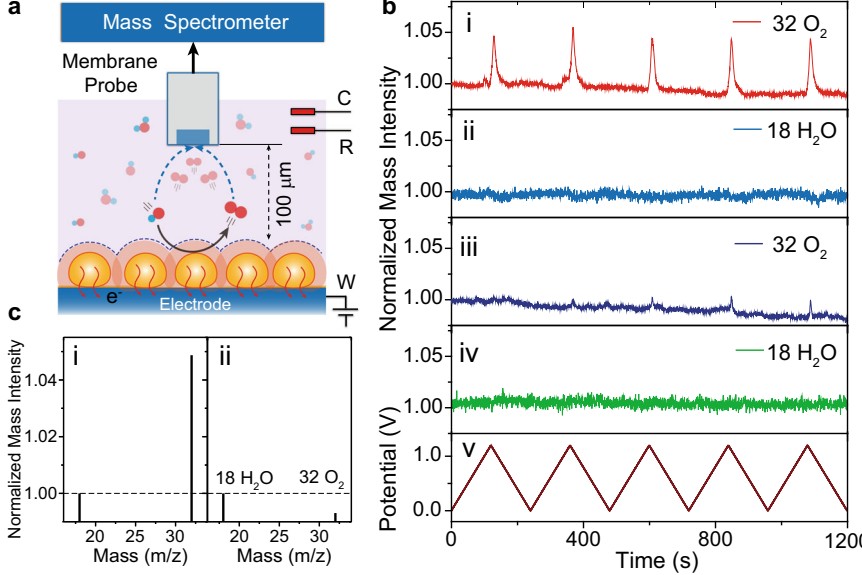

**Fig. 3 Potential-dependent mass spectrometry of dissolved oxygen. a** Schematic of in situ differential electrochemical mass spectrometry. **b** Dynamic mass intensity of O$_2$ and H$_2$O analyzed by in situ DEMS during cyclic voltammetry in the potential range from 0 to 1.10 V on working electrode Cu/glass carbon disc (i, ii) and glass carbon disc (iii) and (iv), respectively, in a home-made micro-electrochemical cell in 0.1 M KOH solution, at a scan rate of 0.01 V s$^{-1}$. **c** The mass spectra during dynamic potential scan on the working electrode Cu/glass carbon disc (i) and glass carbon disc (ii).

(>0.50 V, Fig. 2g). Meanwhile, the extinction peak $a$ assigned to the electrogenerated Cu oxides and hydroxides showed a distinct potential-dependent relationship and remained stable in intensity with increasing anodic polarization, indicating an evident accumulation of dissolved oxygen at the EEIs under OER condition.

We further sought a mass spectrometry analysis to provide direct molecular evidence to solidify that dynamic extinction spectra variations were the results of an electrochemically generated oxygen. Mass spectrometry coupled with electrospray ionization was adopted to in situ analyze oxygen and water (Fig. 3). The oxygen ($O_2$, 32 $m/z$) was captured in the negative mode with predominant intensity than the internal reference mass intensity ($H_2O$, 18 $m/z$). No other mass signals were observed. Notably, the remarkable periodic enhancement was ca. seven times of the oxygen mass intensity than that on bare glass carbon electrode during consecutive anodic polarization from 0 to 1.10 V, providing a direct molecular evidence for the presence of oxygen during electrochemical gas evolution reaction.

**Kinetic analysis of multistage oxygen diffusion at the EEI**. The dynamic extinction spectral evidence for oxygen evolution exhibited different accumulation or dissipation rate of dissolved oxygen on the EEIs and indicated the presence of multistage oxygen diffusion. We next monitored the extinction voltammetry under different potential scan rate, to effectively identify the diffusion of dissolved oxygen (Fig. 4a, b).

Kinetic measurements at different potential scan rate exhibited a square root of scan rate-dependent relationship for peak $b$ and $c$ (Fig. 4a, inset), indicating the oxygen evolution at the Cu NP electrode was restrained by the removal of gas products ($O_2$) from the EEIs rather than the supply of reactants ($OH^-$) from bulk electrolyte under strong alkaline conditions[46,47]. This accumulating inhibition effect originated from the lower solubility and diffusion coefficient of dissolved oxygen, which was 87 and 2.8 times less than that of hydroxide, respectively[48]. Similar oxygen transport-induced accumulating inhibition effect on OER was also observed at single $CoFe_2O_4$ NPs[49]. By increasing the scan rate, the resulting oxygen concentration profile change led to a linear increase of the premature oxygen accumulation rate $k_1$, diffusion rate $k_2$, and nonlinear increasing of accelerated oxygen dissipation rate $k_3$ (Fig. 4c, d). By increasing the oxygen accumulation rate $k_1$, the resulting oxygen concentration profile changes; therefore, the electrode interface will experience various polarization effects that induce inhibition and corresponding deterioration in oxygen evolution performance. In contrast, decreasing the oxygen accumulation rate at a low scan rate will effectively relieve the influence of accumulation of oxygen on the catalytic performance and explain the increased oxygen concentration at the electrode surface. These findings showed that the oxygen accumulation rate $k_1$ was critical in studies that attempted to identify oxygen accumulating inhibition effect on oxygen evolution performance. When increasing the scan rate, a low oxygen concentration at the EEIs was observed due to the severe inhibition on the oxygen evolution induced by the fast oxygen accumulation rate. Despite the low dissolved oxygen concentration on the EEIs, it contributed to the relative fast diffusion rate $k_2$ during the sluggish oxygen diffusion region, in which insufficient anodic polarization cannot steadily drive the oxygen evolution and thus is dominated by the spontaneously dissolved oxygen diffusion. It is noteworthy that a sharp attenuation of dissolved oxygen was observed as enhancing of the cathodic polarization, indicating the presence of accelerated oxygen dissipation near the electrode surface.

The nonlinear increasing of the oxygen dissipation rate $k_3$ as the increasing of scan rate was explained by the enhanced oxygen reduction rate with gradually increasing coverage of the catalytic $Cu(OH)_2$ layer or submonolayer $Cu_2O$ surface. The possible electron relay between adjacent copper sites, $Cu_2O$ and $Cu(OH)_2$, promoted the electron transfer to adsorbed oxygen on the catalytic sites, which was responsible to the enhanced oxygen reduction rate[50]. The ability to visualize these oxygen evolution profiles was a direct result of our plasmonic-enhanced platform that allows in situ monitoring gas evolution at the EEIs under various polarization, which prevented the artificial interferences in ex situ measurements. These results suggested that the dissolved oxygen could be adsorbed on the electrode interface by non-covalent van der Waals interactions, leading to the blockage of active sites and the restraint of charge transfer. This multistage diffusion thus revealed how oxygen accumulation inhibited the oxygen evolution performance.

To further explore the mass transfer of dissolved oxygen and charge transfer at the EEIs, finite element method was performed by combining of Bulter–Volmer equation and Fick's second law (Fig. 4e, f)[51]. The electrocatalytic performances with charging correction decreased as the potential scan rate increased, exhibiting a similar trend with the extinction measurements (Supplementary Figs. 18 and 29). Meanwhile, the numerically simulated oxygen evolution profile at various scan rates showed consistent oxygen evolution and diffusion behavior compared with in situ extinction voltammetry, and confirmed the capability for the extinction voltammetry in monitoring oxygen evolution (Fig. 4e). Moreover, concentration contour maps of the oxygen diffusion layer revealed that the diffusion region of dissolved oxygen was more inclined to being confined at the electrode surface with a shorter diffusion length (150 μm) at a higher scan rate (0.10 V s$^{-1}$) and the thickness of diffusion region increased with decreasing scan rate (Supplementary Fig. 29). Notably, the extinction intensity at peak $b$ showed a good agreement with the simulated oxygen concentration at the EEIs as a square root of the scan rate (Fig. 4f). These results further demonstrated the effectivity of plasmonic single-wavelength strategy in revealing details on the interfacial dynamics directed toward oxygen evolution and further supported that mass transport of dissolved oxygen imposed restriction effects on OER. We proposed that the confinement of dissolved oxygen at the EEIs was responsible for impeding oxygen evolution, leading to a relatively higher oxygen concentration polarization as well as conjectural local hetero-geneous supersaturated regions readily for oxygen bubble nucleation and growth.

**Improving OER performance through programmable potential scan**. Gradual enrichment of dissolved oxygen leads to an increased concentration polarization and a blockage of interfacial active sites at the electrode surface. The precise recognition of multistage oxygen diffusion enables us to accelerate the accu-mulated dissolved oxygen to be removed from the electrode surface without forced convection, further alleviating the dis-solved oxygen concentration polarization and improving elec-trocatalytic performance. We thereby established a programmable potential scan strategy to remove the adsorbed interfacial dis-solved oxygen species (Fig. 5a), to effectively eliminate these adverse effects. In brief, the programmable cathodic polarization facilitated the reduction of nonspecific adsorption of hydrated oxygen located in the outer Helmholtz plane and increased the accessibility of interfacial hydroxide[52]. With the ability of iden-tifying spontaneous diffusion (0.67 to −0.35 V) and accelerated oxygen dissipation (−0.35 to −0.63 V) region, reduction potential multi-pulse at −0.50 V, and at −0.30 and −0.10 V were chosen as

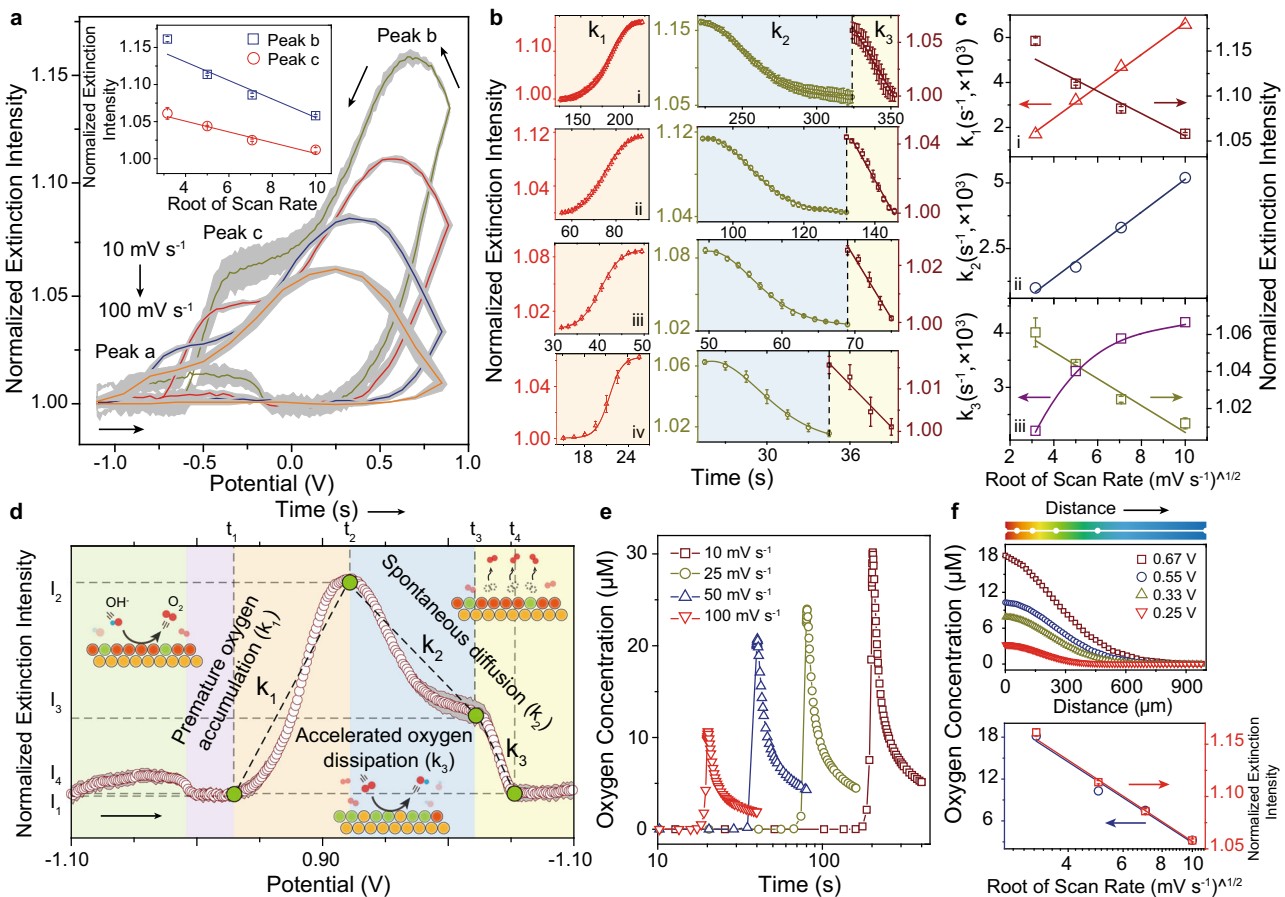

**Fig. 4 Kinetic analysis of multistage oxygen diffusion at the EEIs. a** The scan rate-dependent behavior for extinction voltammetry. Inset shows the relationship between extension intensity (peak *b* and *c*) and the square root of the scan rate. The dark region represents the SE during consecutive five scans. **b** The extinction intensity evolution during oxygen diffusion and oxygen reduction region at various scan rates 0.01, 0.025, 0.05, and 0.10 V s$^{-1}$ (i–iv). **c** The oxygen accumulation rate $k_1$ compared with peak *b* (i), the oxygen diffusion rate $k_2$ (ii), and oxygen reduction rate $k_3$ compared with peak *c* as a function of the square root of the scan rate (iii). Error bar in **b** and **c** represents the SE during consecutive five scans. **d** Schematic of apparent oxygen accumulation rate $k_1$ ($k_1 = | I_2 - I_1 | (t_2 - t_1)^{-1}$), oxygen diffusion rate $k_2$ ($k_2 = | I_3 - I_2 | (t_3 - t_2)^{-1}$), and dissipation rate $k_3$ ($k_3 = | I_4 - I_3 | (t_4 - t_3)^{-1}$) during dynamic potential scan. Inset shows the interfacial reactions. **e** Numerical simulation of oxygen concentration evolution on the electrode surface at various potential scan rate (0.01, 0.025, 0.05, and 0.10 V s$^{-1}$). **f** Numerical simulation of concentration profiles at various polarization potential corresponding to the peak potential (peak *b*) at various scan rates 0.01, 0.025, 0.05, and 0.10 V s$^{-1}$, respectively (top panel). The comparison between the dependence of oxygen concentration at the EEIs and extinction intensity (peak *b*) on the square root of the scan rate (bottom panel).

control potentials to confirm the effectiveness of proposed concept. The overpotential negatively shifted 24 mV under reduction potential multi-pulse (−0.50 V) compared to the positive shift ca. 12 mV at both −0.30 and −0.10 V, indicating the effective alleviation of polarization potential by removing the adsorption dissolve oxygen species (Fig. 5b and Supplementary Fig. 30). Notably, the prominent enhancement 24.1% of current density at 0.70 V under reduction potential multi-pulse (−0.50 V) in contrast to the attenuation 11.8% and 36.7% at −0.30 and −0.10 V, respectively, originated from the increased exposure of interfacial adsorbed hydroxides, a consequence of removing dissolved oxygen from the electrode surface (Fig. 5c and Supplementary Fig. 31). These results confirmed the validity of the electric field-induced elimination of interfacial dissolved oxygen concept on promoting OER performance and suggested a general principle for the improvement of fuel cell reactions at the EEIs.

## Discussion

In summary, we real-time monitored oxygen diffusion at the EEIs by using in situ extinction spectroelectrochemistry and mass spectrometry. The combination of in situ techniques not only provided an unprecedented understanding toward the dynamic behavior of catalytic interface without the interference of non-Faradaic effects but also revealed the oxygen accumulation at the EEIs as a limiting factor on OER and corroborated the presence of multistage diffusion behavior of dissolved oxygen, which cannot be precisely identified in conventional electrochemical methods. The information derived from the present system could be easily transfered to other gas evolving electrochemical reactions, such as $H_2$, $CO_2$, and $N_2$ involving reactions through more complicated catalytic interfacial chemistry. Our strategy can be further generalized to readily evaluate the non-plasmonic electrocatalysts performance by coupling advanced plasmonic antenna, thus allowing efficient development of desirable catalysts and leading to next-generation functional energy device.

## Methods

**Electrochemical measurements**. A CHI660E electrochemical station (CH Instruments, Shanghai, China) was used to validate occurring of gas evolution reaction in a micro-electrochemical cell. Electrochemical measurements were performed in a conventional one-compartment cell in conjunction with a standard three-electrode system and the micro-electrochemical cell, respectively. In the conventional cell, the electrochemical seed-mediated growth of Cu NPs on the ITO electrode was performed according to the above procedures. Linear sweep voltammetry was performed at a scan rate of 0.01 V s$^{-1}$ to qualitatively evaluate the

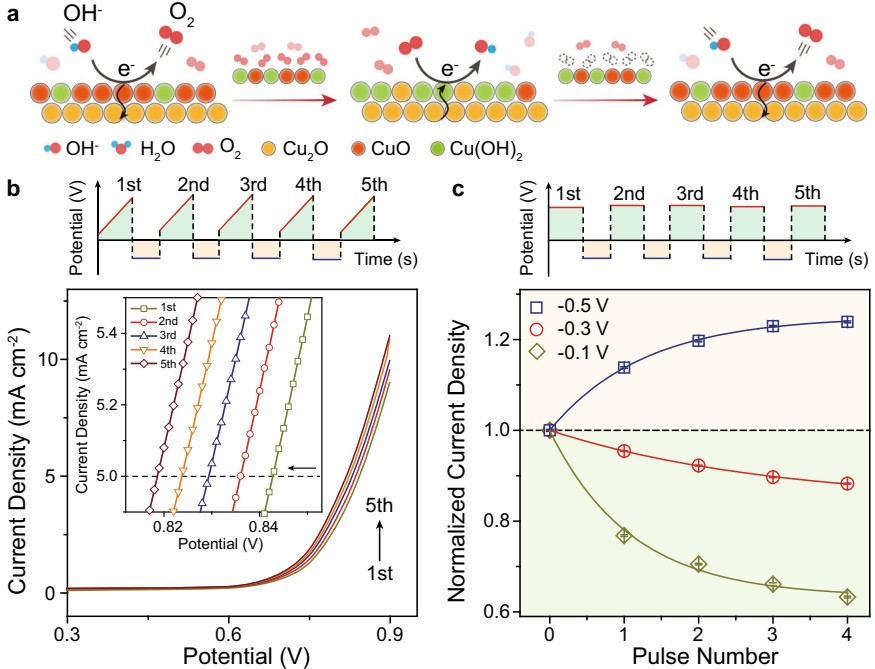

**Fig. 5 Programmable potential scan strategy to eliminate accumulated interfacial dissolved oxygen at the EEIs. a** Schematic illustration of reduction potential pulse accelerating interfacial oxygen species removed from the electrode surface. **b** Linear sweep voltammetry curves of Cu electrode combined with reduction potential multi-pulse (−0.50 V) in 0.10 M KOH solution at a scan rate of 0.01 V s⁻¹. Inset shows cathodic shift of the oxygen evolution potential as consecutive reduction potential pulses (−0.50 V). **c** Mean chronoamperometry current density at 0.70 V as a function of reduction potential multi-pulse (−0.50, −0.30, and −0.10 V) normalized by the value of the original current density (first scan). Error bar represents the SE during consecutive five scans. Schematic of the programmed potential scan strategy shown in the upper panel in **b** and **c**, respectively.

catalytic activity for oxygen evolution and define the onset potential of surface gas evolution.

Electrochemical measurements coupled with in situ extinction spectroscopy were performed in a home-made three-electrode micro-electrochemical cell equipped with a Pt mesh counter electrode (CE) and a Pt wire QRE. The extinction spectra were recorded in a spectral range from 390 to 1000 nm during potentiostatic polarization measurements. Electrochemical extinction measurements were conducted at steady-state conditions and the as-prepared Cu NPs/ITO electrode was allowed to equilibrate at each potential for at least 2 min prior to extinction spectral acquisition. A total of five scans were averaged for one spectrum, which required an accumulation time of 90 s. We began by applying a constant potential of −1.10 V (vs. Pt) and measuring the extinction spectrum over the course of 3.5 min, then applied a more positive potential and re-measured the extinction spectrum for another 3.5 min, continuing in a stepwise manner up to 0.90 V. The evolution of the extinction spectra was tracked as a function of potential.

Kinetic experiments for in situ probing the OER were performed by monitoring the intensity located at extinction peak (recognized from the extinction spectra of each Cu NPs/ITO electrode) over time as a particular voltage in a stepped chronoamperometric mode. On the basis of cyclic voltammetry curves in a 0.1 M KNO₃ solution containing 5.00 mM K₃[Fe(CN)₆] (Supplementary Fig. 1), the $\Delta E_0^{1/2}$ was calculated as 0.286 V between using Pt QRE and saturated calomel electrode. In the micro-electrochemical cell, real-time monitoring of gas evolution reaction during cyclic voltammetry in potential range from −1.10 to 0.90 V (vs. Pt) at different scan rates was investigated by in situ extinction electrochemical measurements. The extinction spectra assigned to fluctuations of the EEI on the Cu NPs/ITO electrode during dynamic potential scan were collected synchronously with corresponding electrochemical information. The electrolyte was bubbled with N₂ gas for 20 min prior to ejection into the micro-electrochemical cell.

Rotating ring-disk electrode (RRDE) electrochemical measurements were performed using either a bipotentiostat model CHI760D (CH Instruments, Shanghai, China) and a Pine model AFCBP1 bipotentiostat (RRDE). During RRDE cyclic voltammogram measurements, the working electrode used a fixed-disk RRDE with glass carbon disk OD 5.61 mm, ring OD 7.92 mm, and ring ID 6.25 mm, and polytetrafluoroethylene (PTFE) shroud 15.0 mm. The apparent RRDE collection efficiency (37.0% at 1600 r.p.m.) was estimated from the ring and disk limiting current ratios in 0.10 M KOH and 5 mM K₃[Fe(CN)₆] solution. The various collection potential (−0.10, −0.50, −0.70, and −0.90 V vs. Pt) were evaluated in RRDE experiments for oxygen reduction reaction. The fabrication of Cu NPs/GC electrode follows similar procedures illustrated in the above section. The CE and QRE in all RRDE experiments were both Pt wire electrodes. The

resistance values estimated from electrochemical impedance measurements ranged from 25 to 36 Ω. All experiments were performed at room temperature 25 ± 2 °C.

**Instruments**. AFM measurements were performed on a Bioscope Resolve AFM (Bruker Corp., USA) under tapping mode with a scan rate of 1.0 Hz frequency. AFM images with a resolution of 512 × 512 pixels were acquired using silicon cantilevers with a normal spring constant of 0.35 N m⁻¹. All recorded images were treated with "Flatten" function prior to analysis in NanoScope Analysis software (version 1.60). Ultraviolet-visible spectra were recorded on a UV-2600 spectrophotometer (Shimadzu, Japan).

ToF-SIMS measurements were performed using a ToF-SIMS V (ION-TOF GmbH, Münster, Germany) mass spectrometer equipped with a time-of-flight analyzer of a reflectron type. In brief, a 30 keV Bi₃⁺ primary ion beam was used at 10 kHz frequency with a pulsed beam current of 0.36 pA and it was focused to be ~450 nm diameter in fast imaging mode. All ion image resolution was 512 × 512 pixels. The positive and negative mass spectra were calibrated by CH₃⁺, H₃O⁺, and NH₄⁺ and C⁻, OH⁻, and C₂⁻, respectively. The vacuum pressure in the main chamber during the measurements was below 3 × 10⁻⁹ mbar.

XPS spectra were acquired with the Axis UltraDLD spectrometer (Kratos Analytical, Ltd) with a monochromatic Al Kα source (hν = 1486.6 eV) and a charge neutralization system. Spectra were taken when the vacuum of the analysis chamber was <5 × 10⁻⁹ Torr. The X-ray source power was set to 105 W (15 KV, 7 mA) for spectra acquisition. The pass energy of 160 and 40 eV were used for survey spectra and narrow scan spectra, respectively. Further, the energy step size of 1 and 0.1 eV were chosen for survey spectra and narrow scan spectra, respectively. For a survey spectrum, the binding energy scanning range was set as 0–1200 eV. After the spectra were acquired, the binding energies correction was performed by setting C1s peak of the adventitious carbon on sample surface at 284.8 eV. Relative atomic concentrations of each detected elements were calculated based on the peak areas and relative sensitivity factors provided by the instrument manufacturer.

X-ray diffraction patterns were collected on a Bruker D8-ADVANCE X-ray diffractometer with DteX Ultra 250 detector and copper target using monochromatic Cu $K_\alpha$ radiation (λ = 1 : 5406 Å). The range of diffraction (Bragg) angles were measured within the scan range from 10° to 80° with a step angle of 0.02° and a scan rate of 20° min⁻¹ at 40 KV, 30 mA. The incident Soller slit and length-limiting slit was set as 2.5° and 5 mm, respectively.

Field-emission transmission electron microscope (TEM) images were collected on a JEOL-2100F (Japan) equipped with an energy dispersive X-ray detector and operated at voltage 200 kV. High-resolution TEM (HRTEM) images, selected area electron diffraction patterns, and relative elemental mapping were carried out

during HRTEM measurements. For TEM characterization, the electrochemically deposited Cu NPs were gently scraped by a steel blade from the ITO electrode and subsequently dispersed in anhydrous acetonitrile solution by sonication (20 s). Then, a droplet of the suspension with 20 μL was gently dropped onto a carbon-coated copper grid (300 mesh), allowing the solvent to evaporate prior to imaging.

To in situ record the spectral signals of Cu NPs/ITO electrode during dynamic electrochemical reaction, the halogen lamp and deuterium lamp were served as the light source. We used Czerny-Turner type monochromator (lens 1300 lines mm$^{-1}$) to provide monochromatic light. A photomultiplier tube was used as a detector to collect the optical signals from the EEIs. Electrochemical seed-mediated growth of Cu NPs was adopted to fabricate plasmonic-enhanced system for single-wavelength monitoring. With the enlargement of NP size, the extinction cross-section was increased gradually and reached saturation after deposition of 150 cycles (Fig. 1, shown in main text). Moreover, the as-prepared Cu nanostructures (deposited 150 cycles) exhibit a pronounced CTP mode at 690 nm (Fig. 1e). Therefore, the electrodes obtained by electrochemical deposition for 150 cycles were used for both electrochemical and optical measurements. Here we monitored the temporal extinction trajectory of Cu NPs/ITO electrode at 690 nm during dynamic potential scan.

## Data availability
The data that support the findings of this study are available from the corresponding author upon reasonable request.

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

## Acknowledgements

This work was supported by the National Natural Science Foundation of China (21675166, 21874046, and 21902050), National Postdoctoral Program for Innovative Talents (BX20190118), Shanghai Post-Doctoral Excellence Program (2019220), Shanghai Municipal Commission for Science and Technology (19JC1411800), the Fundamental Research Funds for the Central Universities, and the original innovation project from 0 to 1 of the Basic Frontier Scientific Research Program of the Chinese Academy of Sciences (ZDBS-LY-JSC010).

## Author contributions

J.-G.W. and D.L. designed research. J.-G.W. and D.L. performed research. J.-G.W., L.L., Y.S., L.S., Z.Y., R.H., J.X., Y.T., and D.L. analyzed data. J.-G.W. and D.L. wrote the paper.

## Competing interests

The authors declare no competing interests.
