## [Peer Review File · Nature Communications]

Reviewer #1 (Remarks to the Author):

In this paper, the authors demonstrated the multistage oxygen diffusion and accumulating inhibition effect in oxygen evolution reactions through in-situ plasmon extinction tracking. The analyses for multistep oxygen diffusion behaviors seem to be interesting. However, the experimental conditions as well as the interpretation in this paper could be carefully considered in addition to electrochemically fabricating copper nanoparticles. The detailed revisions are required as follows:

1. The electrodeposited copper nanoparticles showed totally a random distribution (Figure 1b and Figure S6). However, the authors simulated charge transfer plasmon (CTP) modes of the copper particles as dimers according to their growth. The copper samples in this paper have a considerable variety of particle geometries and inter-particle arrangements. If the application of the CTP modes is not essential, the authors should simulate a single-particle system using the sparse samples.

2. In Figure S7e, the resulting copper nanoparticles had significant irregular shapes rather than spherical. This may cause a significant difference in the local electric double layer, affecting their plasmonic characteristics and oxygen diffusion behavior at the same time (J. Phys. Chem. Lett. 2014, 5, 4331-4335). Could the electrochemical deposition conditions be optimized to produce the copper nanoparticles with a relatively uniform spherical shape?

3. Crystal growth in electrodeposition fairly differs from that by thermodynamic behavior in colloidal systems. Therefore, Ostwald ripening in Figure S6 should be changed to Ostwald ripening-like.

4. The authors characterized the copper phases using XRD data. However, in Figure 1b and Figure S6b, the cube-shaped domains covered a significant portion of the entire surface, which were presumably composed of cuprous oxide rather than Cu(0). In my opinion, it seems that the electrodeposited copper nanoparticles also have a large portion of oxide domains. The plasmon changes can be interpreted in case that a clear analysis of the copper phases is conducted using XPS. Additionally, I wonder that the thick layer formed on the particle surface in Figure S7c indicates the presence of oxide species.

5. How was the inter-particle distance of 26 nm measured, and why was the Au layer in Figure S8a marked? Furthermore, I wonder that the modeling was conducted only with copper domains without oxide layers. If the layer was applied, how was the thickness characterized?

6. In Figure 2e, as the enhancement of anodic polarization, the change of plasmon extinction by the copper nanoparticles was clearly observed, and the three peaks, a,b,c, were explained by both the oxidation of copper and the diffusion of dissolved oxygen with two different modes. However, the explanation of the stable region between peaks a and b is unclear. With increasing anodic potential, the copper nanoparticles may be continuously oxidized and even dissolved, resulting in extinction changes. Therefore, AFM or SEM images to confirm the size and morphology of the particles after peak a and c should be presented with the clear analysis.

7. In general, the oxidation of copper nanoparticles decreases the extinction intensity (Chem. Commun. 2012, 48, 1511-1513). Was the increase in extinction efficiency at the peak a only due to the growth of copper hydroxide domains? If so, with an increase in anodic potential, the plasmon change can be also caused in peak b for a similar reason as in peak a. Therefore, an experimental analysis for the oxide/hydroxide ratio for peak a,b, and the correlation for plasmon changes should

be conducted. The crystal structures of copper hydroxide are quite different from copper oxide, so it should be analyzed through XRD analysis.

8. The reduction in the refractive index of the environment contributes to a decrease in plasmon efficiency and a slight increase in resonance energy. It is also confirmed in equations (5-4) and (5-5) of supporting materials. However, in this experiment, a decrease in the refractive index by oxygen evolution gave rise to an increase in intensity at peak b. Could you comment on the origin of this result and Figure S21d?

Reviewer #2 (Remarks to the Author):

In this manuscript Wang et al report a method of tracking the oxygen evolution reaction dynamics at the electrode-electrolyte interfaces relevant for water-splitting through a direct plasmonic sensing in Cu NPs. With the method they are able to assign and distinguish different processes that occur and later use this knowledge to rationally design a scanning strategy to improve the current density in an OER by 20%.

I found the work and concept to be intriguing which really demonstrates the superiority of a plasmonic sensing to disentangle a complex process. I also found the work to be highly multidisciplinary and will attract readers from multiple fields. I also commend the number of different analysis/experiments done to support their findings.

Having said that, unfortunately I found that the manuscript in its current state to be far from suitable for publication in Nature Communications, with reasons I will detail later below. However, I personally would be interested to see the manuscript be improved since I think it has a high potential.

Questions/recommendations:

1. Page 3: "Gas evolution leads to an undesirable blockage.." What gas? What kind of evolution? How specifically does the blockage happen?
2. Page 3: "at various polarization." Polarization is often mentioned but it is never explain what polarization is this. See also Page 5 "severe polarization effect".
3. Page 4: "advanced plasmonic nanostructures" What does advanced here refer to? Any metal nanostructures whether they are advanced or not still support plasmonics.
4. Page 4: "environment refractive sensitive plasmonic" The authors often use otherwise common terms in the field with their own style. This makes me confuse whether or not I am thinking what the authors are, or they are actually entirely different things. In this case I believe they meant to be "refractive-index sensitive"? Other cases include "near-plasmonic enhancement". I guess this was supposed to be "near field"? Please pay high attention to these well-known terms.
5. Page 7: Is CTP analogy rightly assigned to the studied system here? CTP is formed through a conductive junction between 2 particles but here instead the connection was through the metallic base and therefore not really a junction.
6. Figure 1: One of the major flaws in the manuscript is how they model/simulate their particles. From the schematic and AFM (panel c and d) authors always draw their particles as a dimer with a certain gap distance. Hence all of the simulation throughout the manuscript use this dimer model. However looking at the Figure 1b this is not at all the case. Dimer was just a few percentage of the overall sample. The rest are randomly comprised of arrays of random particle, sometime even a formation of "chain-like" structures. I suggest authors to take a major change to this. Probably get a

statistics on how many actually form dimer/trimer/etc, what are the average particle-to-particle distances, etc. Otherwise all of the simulation lose their credibility in my opinion.

7. Figure 1f: which one is top and which one is side? They need to be clearly labeled. Also in both cases they look round? Shouldnt the one from the side look hemispherical?

8. Page 8: "we observed a remarkable increase..." and the rest of the paragraph. I cant really see what the authors described here in the data. They way Figure 2b is plotted does not help as well.

9. Page 8: "These indicate that the Cu oxide and..." What is the base for this statement? There is no reference and there is data/simulation by the authors to support this.

10. Figure 2c is unclear and hard to understand. What are the colors?

11. On using the single wavelength extinction spectroscopy, I did not find anywhere about details regarding the experiment. What wavelength did the authors used? What was the reason on using that wavelength? Have comparison to other wavelengths been conducted?

12. Page 10: On the five typical potential regions (Figure 2e), it is unclear whether these 5 regions are well-known from the literature or it is derived from the authors observation? Because if it is from the former, I dont see any references. This is important because later in the paragraph they assign the change of the optical signal to different mechanisms that occur based on these 5 regions.

13. Related to point above, the authors then rationalized the increase/decrease in the extinction to process occurring at the interface. One example, in Region 1 the phase growth within an oxide layer is ascribed to the increase of extinction intensity. For all of these I think the authors should try to verify this through their FDTD. Also, again, depending on which wavelength the extinction is taken with respect to the position of the LSPR peak, the response can take form in the increase or decrease. As such all of these correlation between the extinction intensity and processes occurring at the interface are premature.

14. Page 12: Where do the refractive index of 2.9, 2.72 and 1.71 for different Cu state come from? Literature of authors' own experiments?

15. Page 12: Authors argue that the dissolved oxygen accumulated at the electrode to cause reduction of local RI, based on their simulation shown in Fig. S21. The problem here is that the simulation condition is unclear. Instead of technical schematic of how the structures were simulated, there is only an artistic figure of the system. How did the authors simulate the gradient of the O₂ concentration? What is the input for their refractive index? This part should be significantly amended.

16. Page 13: "obsolete potential-dependent relationship". What is the context of the obsolete here?

17. Page 14: "rather than other gas, for example, air". Air is composed of 20% O₂, so in this context air and oxygen are the same. Authors should trace with their MS other relevant gases that occur in the reaction, not just H₂O like the show in Fig. 3.

18. Figure 5a: Please add legends for all the symbols.

19. Lastly, the manuscript is unfortunately not great in terms of its language. It is riddled with grammatical errors and inappropriate terms that make it very difficult to grasp the idea efficiently. One prime example, even the title is not grammatically correct. I strongly recommend the authors to take a significant (professional) proofreading for all of the texts (main text and SI).

Thank you very much and good luck.

Reviewer #3 (Remarks to the Author):

In this manuscript, the authors proposed an in situ spectroelectrochemical method to uncover the

multi-stage diffusion of oxygen molecules during OER catalyzed by copper nanoparticles. Guided by the inhibition effect of oxygen accumulation they discovered, they further improved the OER performance through a rationally-designed potential scan sequence. Their spectroelectrochemical method relied on the sensitive dependence of the plasmonic spectrum of a semi-transparent ITO substrate (which was pre-electrodeposited with copper nanoparticles) on the refractive index of electroactive surface (copper nanoparticles here) as well as the surrounding medium (oxygen molecules in this case).

The general impression from this reviewer is that, the study was nicely designed and conducted. In addition to the quite impressive data quality and in-depth discussion, this reviewer particularly likes the last part on the improved OER performance. While many studies in the field stopped at the mechanism clarification, this work sets a rare example showing how mechanism advancements indeed helped the performance improvements. This reviewer thus supports the publication of this study in Nature Commun if the authors can address the following points.

1. While this study relied on the extinction spectrum (and intensity at single wavelength), experimental descriptions on the optical/spectral apparatus are almost completely missing (Fig. 1a). The light source (LED? Lamp? White Laser?), the spectrometer, the detector (DAD? PD?) must be specified. Whether or not an objective/lens was used in the light path to collect the light?
2. In Fig. S7c, HR-TEM image was given. It was unclear how the authors prepared the sample (transfer from ITO substrate to TEM grid)? A ~ 1 nm-thick shell was identified from the image. Please explain.
3. In Fig. 1b, it is recommended to display potential-dependent spectra in a complete cycle from -1 to 1 to -1 again.
4. Is it possible for the authors to discuss on the sensing distance of Cu nanostructures to report the oxygen amount/concentration? It is clear that only the oxygen molecules nearby the nanostructure surface could contribute the extinction. But how far?
5. In Fig. 2e, it remains difficult for this reviewer to understand why the onset potential in extinction signal was so much earlier than that in current signal. The authors explained that current signal was consisted of both Faradaic and nonFaradaic components. It is true. But given the quite small non-Faradaic charging background and the significant extinction signal in the early stage in region III, there could be some other reasons behind the mismatch. This point is more concerning when looking at the MS-data in Fig 3b, red curve. The onset potential in mass signal is not consistent with that in the extinction signal, while they should be consistent because they both indicate the amount of product (oxygen molecules).

Reviewer: 1

In this paper, the authors demonstrated the multistage oxygen diffusion and accumulating inhibition effect in oxygen evolution reactions through in-situ plasmon extinction tracking. The analyses for multistep oxygen diffusion behaviors seem to be interesting. However, the experimental conditions as well as the interpretation in this paper could be carefully considered in addition to electrochemically fabricating copper nanoparticles. The detailed revisions are required as follows:

Reply: We are deeply thankful to the reviewer for her/his constructive evaluation and appreciation of this work. According to valuable suggestions, we have made corresponding modifications in our manuscript to address the reviewer's concerns.

1. The electrodeposited copper nanoparticles showed totally a random distribution (Figure 1b and Figure S6). However, the authors simulated charge transfer plasmon (CTP) modes of the copper particles as dimers according to their growth. The copper samples in this paper have a considerable variety of particle geometries and inter-particle arrangements. If the application of the CTP modes is not essential, the authors should simulate a single-particle system using the sparse samples.

Reply: Following the reviewer's suggestion, we simulated a single particle system based on Mie theory (Figure R1). The simulated extinction cross section of single particle exhibits only a slight red shift as a function of particle size. The simulation of dimer-on-film structure, however, were closer to experimental results (Figure S8), suggesting the dimer-on-film structure was more suitable in this study, while only considering single particle cannot fully describe the optical performance of the ensemble system. The simplified dimer-on-film model thus captures the key feature of the ensemble system that enables us to estimate the optical properties of Cu NPs/ITO electrode.

To further highlight the importance of CTP modes, we chose Cu NPs/ITO electrode (5 cycles) with less dimer-on-film structures as a control (Figure R2, a). We compared electrodes with high and low abundance of dimer-on-film structures (Figure R2, c). From in-situ extinction trajectories (Figure R2e), the high density dimer-on-film structures exhibit

ca. 5 times sensitivity enhancement in OER, which significantly facilitates the monitoring of dynamic electrochemical interfacial reactions. Therefore, the CTP mode derived from high coverage density of dimer-on-film structures contributed significantly to the plasmonic monitoring.

Figure R1. The color-code map of normalized extinction (a), scattering (b) and absorption (c) cross section evolutions as a function of Cu NPs size. The representative simulated extinction (d), scattering (e) and absorption (f) cross section of Cu NPs with various diameter (i 35 nm; ii 74 nm; iii 100 nm; iv 164 nm; v 227 nm; vi 234 nm).

Figure R2. (a) AFM image of Cu NPs (deposition 5 cycles) (scale bar: 1 μm). (b) Ex-situ extinction spectra of Cu NPs at 5 cycles. (c) AFM image of Cu NPs (deposition 150 cycles) (scale bar: 1 μm). (d) Ex-situ extinction spectra of Cu NPs at 150 cycles. (e) The consecutive extinction trajectories of (i) a Cu/ITO electrode (150 cycles) and (ii) a bare ITO electrode monitored at 690 nm; (iii) a Cu/ITO electrode (5 cycles) and (iv) a bare ITO electrode monitored at 600 nm under dynamic potential scan from -1.10 to $+0.90$ V at a scan rate 0.01 V s^{-1} (5 cycles). (f and g) The comparison of the maximum spectral intensities at various monitoring wavelength (f 690 nm; g 600 nm).

2. In Figure S7e, the resulting copper nanoparticles had significant irregular shapes rather than spherical. This may cause a significant difference in the local electric double layer, affecting their plasmonic characteristics and oxygen diffusion behavior at the same time (*J. Phys. Chem. Lett.* 2014, 5, 4331-4335). Could the electrochemical deposition conditions be optimized to produce the copper nanoparticles with a relatively uniform spherical shape?

Reply: We agree with this reviewer that single particle study with uniform size could indeed offer new possibilities in reaction monitoring (for example, see pioneer work by Mulvaney's group: *Electrochemical charging of single gold nanorods*, *J. Am. Chem. Soc.* 2009, 131, 14664–14666; *Direct observation of chemical reactions on single gold nanocrystals using surface plasmon spectroscopy*, *Nature Nanotech.*, 2008, 3, 598-602; *Spectroelectrochemistry of silver deposition on single gold nanocrystals*, *J. Phys. Chem. Lett.* 2014, 5, 4331–4335).

However, we would like to emphasize that the present data were derived from ensemble samples that involving various particle sizes and morphologies, and ensemble averaged

results could cover the variations from individual particles.

The difficulties in synthesizing uniform Cu NPs by electrodeposition lies in the multiple applied overpotential that required for the regulating of multiple growing steps. In classical nucleation theory, nucleation of a new solid phase on a heterogeneous interface has to overcome a free energy barrier related to the thermodynamic costs for forming a critical cluster of atoms. During electrodeposition, this nucleation energy barrier can be effectively regulated by changing the local electrochemical supersaturation through adjusting the applied overpotential on the reduction reaction. The electrodeposition, however, involves multiple steps including charge transfer, mass diffusion and interfacial reactions as well as crystallization (*Electrocrystallization: fundamental considerations and application to high current density continuous steel sheet plating. Journal of Applied Electrochemistry 1991, 21, 377-385*). Overpotential is however a complex parameter. Here we referred to nucleation (η_n) and growth (η_g) overpotential for explicit describing Cu electrodeposition (Figure S1). During initial nucleation stage, an essential prerequisite for nucleation of embryos is a sufficient overpotential that surmounting the large free energy barrier (ΔG_n). After nucleation occurs, a lower overpotential is competent in subsequent growth process owing to the lowering free energy barrier (ΔG_g) for the growth of Cu seeds. Because of the lower free energy barrier, it is thermodynamically favorable to introduce a new Cu adatom on a Cu nuclei interface rather than re-nucleation on a non-homologous heterogeneous interface. Taken together the different free energy barrier in both nucleation and growth, an electrochemical seed-mediated strategy was adopted to efficiently adjust the Cu electrodeposition (Figure R3, a and b).

Based on above considerations, we firstly optimized the electrodeposition potential of copper seeds to avoid the interference from hydrogen evolution (Figure S3 and S4, Supporting Information). Then we adjusted the deposition cycles to achieve the tunability of particle size and optical properties (Figure 1 and Figure S5, S6). Both of these efforts could improve the homogeneity of the electrodeposited Cu NPs to avoid the obvious aggregation of particles. Form the large view of SEM images of Cu NPs/ITO electrode (Figure R4), we could see a relative homogeneous distribution of Cu NPs. Furthermore, the explicit extinction

peak of the Cu NPs/ITO electrode corroborates the homogeneity of Cu NPs (Figure 1, main text).

Despite lots of efforts made to improve the homogeneity of nanoparticles, it should be noted that due to the intrinsic limitation of non-template electrochemical methods in the precise control of size and morphology, we cannot achieve more accurate morphology controllability, such as in electron beam evaporation or other photolithography techniques.

Figure R3. (a) Schematic illustration of the free energy evolution during electrodeposition of Cu NPs. The nucleation energy barrier of Cu seeds decreases with increasing of the applied overpotential. (b) Schematic of the electrochemical seed-mediated growth of Cu NPs on ITO electrode. This strategy involves two main steps: (1) a seeding step under enhanced cathodic polarization; (2) a growth step with lower cathodic polarization. The schematic of overpotential and relative current for electrochemical deposition of Cu NPs are shown in top and bottom panel, respectively.

Figure R4. (a-c) Representative SEM images of Cu NPs/ITO electrode (electrodeposition 150 cycles) with different magnifications. A large area view of the electrode is shown to confirm the quasi-homogeneous distribution of Cu NPs (a). (d-e) Representative dimer structure on Cu NPs/ITO electrode. (h-i) Control experiment for bare ITO electrode with different magnifications showing clean background.

3. Crystal growth in electrodeposition fairly differs from that by thermodynamic behavior in colloidal systems. Therefore, Ostwald ripening in Figure S6 should be changed to Ostwald ripening-like.

Reply: We agree that crystal growth in electrodeposition is different from that by thermodynamic behavior in colloidal systems. As suggest by the reviewer, we have modified “*Ostwald ripening*” as “*Ostwald ripening-like*” in Figure S6.

4. The authors characterized the copper phases using XRD data. However, in Figure 1b and Figure S6b, the cube-shaped domains covered a significant portion of the entire surface, which were presumably composed of cuprous oxide rather than Cu(0). In my opinion, it

seems that the electrodeposited copper nanoparticles also have a large portion of oxide domains. The plasmon changes can be interpreted in case that a clear analysis of the copper phases is conducted using XPS. Additionally, I wonder that the thick layer formed on the particle surface in Figure S7c indicates the presence of oxide species.

Reply: (1) Following the reviewer's suggestion, we further performed XPS to identify the interfacial species and to determine the chemical states of the Cu NPs/ITO under various polarization (Figure R5 and R6, Figure R5 was also added in the Supplementary Information as Figure S21). Figure R5 displays XPS spectra of Cu NPs/ITO at various polarized potential and control samples (bare ITO and copper film). The XPS spectra reveals Cu 2p_{3/2} and Cu 2p_{1/2} peaks at 932.5 and 952.0 eV, respectively. In the high-resolution spectra, the Cu 2p_{3/2} at 933.7 eV, 935.1 eV and a series of satellites 943.0 and 940.0 eV are assigned to Cu(II); and the peak at 932.0 eV is correlated to Cu(0) and Cu(I) (*Shape and size control of Cu nanoparticles by tailoring the surface morphologies of TiN-coated electrodes for biosensing applications, Langmuir, 2013, 29, 16025-16033; Oxidation of polycrystalline copper thin films at ambient conditions, J. Phys. Chem. C, 2008, 112, 1101-1108*).

Despite some overlapping, Cu(0) and Cu(I) can still be distinguished from the Cu Auger LMM spectra. As revealed in Figure R5, Cu Auger LMM of Cu film and fresh electrodeposited Cu NPs both exhibited the presence of Cu(I) component located at 570.2 eV, which might be attributed to the fast oxidation of metallic Cu during transferring to the XPS chamber in air (*Improved CO₂ electroreduction performance on plasma-activated Cu catalysts via electrolyte design: halide effect, ACS Catalysis, 2017, 7, 5112-5120; Oxidation of polycrystalline copper thin films at ambient conditions, J. Phys. Chem. C, 2008, 112, 1101-1108*).

The surface species changes during dynamic electrochemical scan was shown in Figure R5 (e-h). After linear potential scan from -1.10 V to -0.36 V in 0.1 M KOH, the amounts of Cu(I) species increased *ca.* 20 %. With further enhancement of anodic polarization at +0.15 V, the emergence of Cu(II) (933.7 eV) and Cu(II) satellite peaks indicated the presence of CuO and Cu(OH)₂ in the passive region (from -0.10 V to +0.20 V). Interestingly, ***comparing the interfacial composition at +0.15 V (passive region) and +0.67 V (peak b),***

negligible change was observed, thus excluding the possible fluctuation of extinction intensity in peak b induced by interfacial phase transitions. Therefore, the corresponding extinction intensity dynamics in peak b was attributed to the oxygen multi-diffusion behavior. In oxygen accelerated region (– 1.10 V), the reduction of oxygen resulted in remarkable adsorption of hydroxyl groups on the surface, that confirmed by the obvious increase of the intensity of Cu(OH)₂ and OH⁻ in O 1s peak (531.9 eV). In summary, the above quasi in-situ XPS analysis reveals detailed phase transitions under various polarization and provides direct chemical information to corroborate our electrochemical experiments.

(2) The shell around the copper nanoparticle (shown in TEM image) was slight surface oxides that also caused by fast oxidation of metallic Cu during the transferring to the TEM chamber in air (*Oxidation of Polycrystalline Copper Thin Films at Ambient Conditions, J. Phys. Chem. C 2008, 112, 4, 1101–1108*). Figure R7d exhibits lattice spacing of the shell region corresponding to Cu₂O(111) (0.24 nm) and Cu(111) (0.21 nm), respectively (*Visualizing the Cu/Cu₂O Interface Transition in Nanoparticles with Environmental Scanning Transmission Electron Microscopy, J. Am. Chem. Soc. 2017, 139, 1, 179–185*). The two spacing of shell (R3) and central regions (R4) are directly related to each other with parallel epitaxy between Cu and Cu₂O *fcc* crystal structure.

High-sensitivity energy-dispersive X-ray spectroscopy (EDS) and scanning transmission electron microscopy (STEM) were introduced to further probe the interface of the Cu NPs (Figure R8). Relatively weak oxygen element was found around Cu NPs, while obvious oxide layer around the Cu NPs was not observed. Moreover, the lattice spacing of 0.21 nm corresponds to the (111) spacing of Cu. Even so, we cannot completely exclude the possibility of fast oxidation of metallic Cu during the transfer in air to the TEM chamber (*Oxidation of Polycrystalline Copper Thin Films at Ambient Conditions, J. Phys. Chem. C 2008, 112, 4, 1101–1108*).

In order to make it more clarification for readership, we added the supplementary sentence “*The slight surface oxides was caused by exposure to air during transport to the TEM.*” in the caption of Figure S7 (Page S16).

Figure R5. Quasi in-situ XPS O 1s spectra, Cu Auger LMM and Cu 2p core level spectra obtained from a bare ITO electrode (a); a copper film (b); Cu NPs/ITO electrode (electrodeposition 150 cycles) (c); Cu NPs/ITO (150 cycles) after linear potential scan from -1.10 V to -0.36 V (d); Cu NPs/ITO (150 cycles) after linear potential scan from -1.10 V to $+0.15$ V (e); Cu NPs/ITO (150 cycles) after linear potential scan from -1.10 V to $+0.90$ V then to $+0.67$ V (f); Cu NPs/ITO (150 cycles) after linear potential scan from -1.10 V to $+0.90$ V then to -0.35 V (g); Cu NPs/ITO (150 cycles) after linear potential scan from -1.10 V to $+0.90$ V then to -1.10 V (h), at a scan rate 0.01 V s^{-1} . Fitted peaks for individual species are shown with solid lines and annotations. O 1s includes the possible species: $\text{Cu}(\text{OH})_2$, OH^- , $\text{O}_{\text{lattice}}$ (oxide lattice atoms). Cu Auger LMM includes the interfacial species: $\text{Cu}(0)$ (568.2 eV); $\text{Cu}(I)$ (570.2 eV); transition state (573.1 eV); Cu 2p includes the interfacial species: $\text{Cu}(0)$ and $\text{Cu}(I)$ (932.5 eV); $\text{Cu}(II)$ (933.7 eV); $\text{Cu}(\text{OH})_2$ (935.1 eV); $\text{Cu}(II)$ satellite peaks (943.0 and 940.0 eV).

Figure R6. XPS spectra of a bare ITO electrode (i); Cu film (ii); Cu NPs/ITO (electrodeposition 150 cycles, Initial state) (iii); Cu NPs/ITO (150 cycles) after linear potential scan from -1.10 V to -0.36 V (iv); Cu NPs/ITO (150 cycles) after linear potential scan from -1.10 V to $+0.15$ V (v); Cu NPs/ITO (150 cycles) after linear potential scan from -1.10 V to $+0.90$ V then to $+0.67$ V (vi); Cu NPs/ITO (150 cycles) after linear potential scan from -1.10 V to $+0.90$ V then to -0.35 V (vii); Cu NPs/ITO (150 cycles) after linear potential scan from -1.10 V to $+0.90$ V then to -1.10 V (viii), at a scan rate 0.01 V s^{-1} .

Figure R7. (a) The enlarged high resolution TEM image shown in Figure S7c and the magnified shell region (b) and central region (c) in Cu NP. (d) Line intensity profiles from the green and red boxes in (b) and (c) with annotations indicating the lattice spacing 0.24 nm corresponding to $\text{Cu}_2\text{O}(111)$, and 0.21 nm corresponding to $\text{Cu}(111)$, respectively. The slight surface oxides were caused by exposure to air during transport to the TEM

Figure R8. Annular bright-field (ABF) STEM (a), medium-angle annular dark-field (MAADF) STEM (b), and high-angle annular dark-field (HAADF) STEM image (c) of Cu NPs (after electrodeposited 150 cycles). (d-f) The EDS element mapping images of Cu NPs corresponding to image (c). The oxygen element shows a random distribution pattern around Cu NP. It can be tentatively attributed to the contribution from air exposure during sample transport. (g) High resolution image of Cu NP and the enlarged square region was shown in (h). (i) Line intensity profiles from the green and red boxes in (h) with annotations indicating the lattice spacing 0.21 nm corresponding to Cu(111), respectively.

5. How was the inter-particle distance of 26 nm measured, and why was the Au layer in Figure S8a marked? Furthermore, I wonder that the modeling was conducted only with copper domains without oxide layers. If the layer was applied, how was the thickness characterized?

Reply: (1) We provided a statistical analysis on the average particle-to-particle distances (Figure R9). According to the definition of gap, only two adjacent NPs with a minimum distance comparable to surrounding particles are considered as a candidate dimer for statistics. From the statistical analysis in Figure R9b, we found a decrease of gap from 479 nm to 55

nm with deposition cycles from 5 to 150 cycles accomplished with a gradual growth of particle size, and a slightly increase of gap from 55 nm to 97 nm with deposition cycles from 150 to 300 cycles caused by the Ostwald ripen-like effect (*Effect of size, coverage, and dispersity on the potential-controlled Ostwald ripening of metal nanoparticles, Langmuir 2019, 35, 16416–16426*). In order to eliminate the contribution from random dispersed particles induced over-evaluation on the gap distance, we adopted a half value of the gap distance (26 nm for Cu NPs/ITO, 150 cycles) in simulation. From the simulation results (Figure S8), similar spectral evolution trend was achieved comparable to experiment measurements (Figure 1e).

Figure R9. (a) Representative AFM images of Cu NPs at different deposition cycles (i 5, ii 50, iii 150 and iv 300 cycles) (scale bar: 1 μm). (b) The corresponding gap distribution (peak location and full width at half maxima, FWHM) of electrodeposited Cu NPs. The minimum gap between two adjacent nanoparticles was chosen for statistical analysis.

(2) Thank the reviewer for careful read and thoughtful comments. It should be “Cu” rather than “Au”. According to the reviewer’s suggestion, we have made corresponding modification in Figure S8.

(3) The thickness of the oxide layer (4 nm) was derived from high-resolution TEM images (Figure 7c). Of note, the Cu_2O growth is accomplished by a phase transition between copper oxide and hydroxide occurred at the electrode surface, which will cause variations of

thickness. Thus, we also simulated the optical response of dimers with different thickness of Cu_2O layer (the thickness of Cu_2O layer was rationally set from 1 nm to 4 nm in the potential range from -1.10 V to -0.36 V; then decrease back to 1 nm in the potential range from -0.36 V to -0.10 V). As shown in Figure 2e (Region I corresponding to peak a), we found the evolution of simulated extinction intensity similarly to experimental result.

The composition of electrode interface suffers great changes during dynamic potential scan, including interfacial composition dynamics, phase transition and surrounding mass transfer. Because of these all complexities, we considered only the contribution from Cu_2O layer and ignored both copper oxide and hydroxide to extinction intensity. We confess that a more complicated model with higher temporal and chemical resolution may help to deeply reveal the interfacial reaction, but it far beyond current capability.

6. In Figure 2e, as the enhancement of anodic polarization, the change of plasmon extinction by the copper nanoparticles was clearly observed, and the three peaks, a,b,c, were explained by both the oxidation of copper and the diffusion of dissolved oxygen with two different modes. However, the explanation of the stable region between peaks a and b is unclear. With increasing anodic potential, the copper nanoparticles may be continuously oxidized and even dissolved, resulting in extinction changes. Therefore, AFM or SEM images to confirm the size and morphology of the particles after peak a and c should be presented with the clear analysis.

Reply: (1) During electrochemical interfacial reaction, we assigned peak a, b and c to the oxidation of copper and the diffusion of dissolved oxygen according to both electrochemical and spectral results. Moreover, we distinguished 5 characteristic regions during dynamic potential scan. These five regions were also derived from both synchronized electrochemical and spectral measurements to explain the dynamic interfacial processes (Figure 2e). These five regions were: (I) active region, where the oxidation of Cu occurs; (II) passive region, where oxygen evolution is suppressed; (III) transpassive region, where the onset oxygen evolution region is controlled by reactive kinetics; (IV) diffusion region, where oxygen diffusion regions is dominated by diffusion kinetics and (V) reductive region, where oxygen

reduction occurs.

At the *interfacial passive region* ($-0.10\text{ V} \sim 0.20\text{ V}$) between peak a and peak b, the formation of a thick and complex Cu oxides and hydroxides passive layer restrains the interfacial oxidation. Thereby, no obvious oxygen evolution was observed (Region ii). The relative stable state at this region could be tentatively attributed to: (1) the insufficient anodic polarization at this region cannot induce oxygen evolution reaction; (2) the interfacial compositions ($\text{CuO}/\text{Cu}(\text{OH})_2$) maintain relative stable (as confirmed by the steady OH/O ratio, Figure R12).

(2) Following the reviewer's valuable suggestion, we compared both morphology and size of the particles at initial state, linear potential scan after peak a (during potential scan from -1.10 to -0.10 V) and linear potential scan after peak c (during potential scan from -1.10 to $+0.90\text{ V}$ then to -1.10 V). From ex situ SEM images (Figure R10), we found that the size and morphology of Cu NPs changed slightly when linear potential scan was used in electrochemical interfacial reaction. Moreover, the stable spectral performance during consecutive potential scan further confirmed the stability of electrodeposited interfaces. These results indicated the dynamic potential scan efficiently avoids remarkable dissolution of soluble copper oxides and hydroxides, while steady-state electrochemical measurements resulted in severe damage of the electrode surface and hampered interfacial processes dissection (Figure 2b). Therefore, dynamic potential scan was used in the present study (Figure 2d).

Figure R10. (a-c) Representative SEM images of Cu NPs/ITO electrode at initial state (left column) after linear potential scan from -1.10 V to -0.10 V (middle column, after peak a) and after linear potential scan from -1.10 V to $+0.90$ then to -1.10 V (right column, after peak c) at a potential scan rate 0.01 V s^{-1} .

7. In general, the oxidation of copper nanoparticles decreases the extinction intensity (Chem. Commun. 2012, 48, 1511-1513). Was the increase in extinction efficiency at the peak a only due to the growth of copper hydroxide domains? If so, with an increase in anodic potential, the plasmon change can be also caused in peak b for a similar reason as in peak a. Therefore, an experimental analysis for the oxide/hydroxide ratio for peak a, b, and the correlation for plasmon changes should be conducted. The crystal structures of copper hydroxide are quite different from copper oxide, so it should be analyzed through XRD analysis.

Reply: As suggested by the reviewer, we first performed XRD analysis to identify copper hydroxide and copper oxide in order to reveal the interfacial oxide/hydroxide ratio during dynamic potential scan. As shown in Figure R11, we found only weak diffraction pattern of Cu(111) at 43.3° (PDF 00-004-0836) and remarkable signals from ITO substrate. The weak Cu signal was caused by the larger penetration depth of X-ray beam than the thickness of the Cu NPs and film structures (*XRD evidence of macroscopic composition inhomogeneities in*

the graphite–lithium electrode, Journal of Power Sources, 2007, 165, 616-619; Plasma-activated copper nanocube catalysts for efficient carbon dioxide electroreduction to hydrocarbons and alcohols, ACS Nano 2017, 11, 5, 4825–4831). Thus, XRD was less efficient to identify the various compositions on the Cu electrode surface.

We next performed Time-of-Flight Secondary Ion Mass Spectrometry (ToF-SIMS) to obtain precise interfacial elements or molecules information. ToF-SIMS is a sensitive surface technique that enables to distinguish interfacial species with the ability to differentiate a proton between two similar species (*Direct molecular evidence of proton transfer and mass dynamics at the electrode–electrolyte interface, J. Phys. Chem. Lett. 2019, 10, 251-258; Real-time mass spectrometric characterization of the solid–electrolyte interphase of a lithium-ion battery. Nature Nanotechnol. 2020, 15, 224-230).*

ToF-SIMS results indicated a distinct fluctuation of interfacial OH/O ratio under various polarization (Figure R12 and R13, these figures were also added in the Supplementary Information as Figure S22 and S23):

- (1) In **peak a**, a gradual OH/O ratio increase accompanied by slight decrease was observed during potential range – 1.10 V to – 0.10 V, which was of the same evolution trend as predict by electrochemical measurements. The dynamic of extinction intensity in peak a could be attributed to interfacial Cu oxides transitions.
- (2) In **passive region** (potential from – 0.10 V to + 0.20 V), the OH/O ratio maintains a steady state, which provides another direct molecular evidence to confirm the interfacial stability.
- (3) In **peak b** (potential from + 0.20 to + 0.90 V then to + 0.67 V), no obvious fluctuation of OH/O ratio was observed. This helps to exclude the possibility that the interfacial oxide transitions lead to remarkable extinction intensity changes. Therefore, the dramatic fluctuation of extinction intensity in peak b can only be correlated with oxygen evolution reaction.

Figure R11. (a) X-ray diffraction patterns of bare ITO electrode (i) and Cu NPs/ITO electrode at initial state (ii); after linear potential scan from -1.10 V to -0.36 V corresponding to peak a (iii); and after linear potential scan from -1.10 V to $+0.90$ V then to $+0.67$ V corresponding to peak b (iv). (b) The enlarged view of the Cu(111) pattern. The 2θ peak at 43.3° is corresponded to the (111) planes of Cu (PDF 00-004-0836).

Figure R12. (a) Schematic of quasi in-situ ToF-SIMS monitoring the interfacial OH/O ratio at the Cu NPs/ITO electrode surface. The high-energy primary ion beam Bi_3^+ bombards the electrode surface causes emission of secondary ions, which can be accelerated and captured by a specific polarized electric field and then transferred to a time-of-flight mass analyzer. It can provide specific compound identification of elements and molecules on a surface. (b) The representative dynamic extinction trajectory (\circ), current curve ($-$) and normalized OH/O ratio (\square) for OER on a plasmonic Cu NPs electrode scanned from -1.10 to $+0.90$ V at a scan rate 0.01 V s^{-1} . The dash line is used to guide the eye for comparison the OH/O ratio. The detailed negative mass spectra of Cu NPs/ITO electrode at various potential are shown in Figure R13.

Figure R13. Negative ion mass spectra of Cu NPs/ITO electrode at various polarized potential: (a1) the initial state of Cu NPs/ITO electrode; (b1) after linear potential scan from -1.10 V to -0.80 V; (c1) after linear potential scan from -1.10 V to -0.36 V; (d1) after linear potential scan from -1.10 V to -0.10 V; (e1) after linear potential scan from -1.10 V to $+0.15$ V; (f1) after linear potential scan from -1.10 V to $+0.70$ V; (g1) after linear potential scan from -1.10 V to $+0.90$ V then to $+0.67$ V; (h1) after linear potential scan from -1.10 V to $+0.90$ V then to $+0.30$ V; (i1) after linear potential scan from -1.10 V to $+0.90$ V then to -0.35 V; (j1) after linear potential scan from -1.10 V to $+0.90$ V then to -0.63 V; (k1) after linear potential scan from -1.10 V to $+0.90$ V then to -1.10 V. (a2-k2) and (a3-k3) are the enlarged view of O^- and OH^- corresponding to various potential (a1-k1), respectively. The OH^-/O^- ratio was calculated by comparison the integral area between OH^- and O^- .

8. The reduction in the refractive index of the environment contributes to a decrease in plasmon efficiency and a slight increase in resonance energy. It is also confirmed in equations (5-4) and (5-5) of supporting materials. However, in this experiment, a decrease in the refractive index by oxygen evolution gave rise to an increase in intensity at peak b. Could you comment on the origin of this result and Figure S21d?

Reply: (1) We agree that a decrease in the refractive index will lead to a concomitant

increased extinction intensity. As in the example of peak a, while the potential was scanned from -1.10 to -0.36 V, a temporal evolution from Cu_2O (refractive index 2.90) to $\text{CuO}/\text{Cu}(\text{OH})_2$ (2.72/1.71) phases resulted in an increased extinction intensity. In contrast, after -0.36 V, potential-driven composition evolution of Cu oxides and hydroxides diminished the OH/O ratio by releasing H, which led to an increase of the refractive index and a rapid attenuation of the extinction intensity.

However, in peak b, possible interfacial oxide transition was excluded by the stable OH/O ratio, and the extinction intensity only be correlated with oxygen evolution reaction. Note that the fluctuation of environmental dielectric constant induced by interfacial electrochemical processes, such as interfacial mass transfer, ionic strength dynamics and interfacial species solvation under an electric field, is more complicated than our qualitatively simplified assumption. A more quantitative description of interfacial reactions needs more concrete modeling and investigation.

(2) The simulation of Figure S21d was carried out with different environmental refractive index. Upon the evolution of oxygen, the refractive index of electrolyte (1.33) located at interfacial region could gradually decrease due to the accumulation of oxygen molecules (refractive index: 1.00, *Precision refractive index measurements of air, N₂, O₂, Ar, and CO₂ with a frequency comb. Applied optics, 2008, 47, 3143-3151*). Because of the limitation in precise identification of absolute oxygen concentration close to the electrode surface, it is very hard and impossible to acquire the dynamic hybrid refractive index of interfacial electrolyte (including solvated oxygen) during dynamic potential scan. To qualitatively evaluate the influence of oxygen accumulation, we rationally set the refractive index of environment ranging from 1.33 to 1.06 in simulating the dynamic environment in potential range from $+0.20$ V to $+0.90$ V then to $+0.67$ V (Region III corresponding to 130 s to 220 s in time scale). To further evaluate subsequent oxygen spontaneous and accelerated diffusion processes, the refractive index of environment was then rationally set from 1.06 to 1.20 in potential range from $+0.67$ V to -0.35 V (Region IV corresponding to 220 s to 319 s in time scale) and from 1.20 to 1.33 in potential range from -0.35 V to -0.63 V (Region V corresponding to 319 s to 365 s in time scale), respectively. While the evolution of the extinction intensity showed a

similar trend compared to the experimental results, which further confirmed the reasonability of our proposed model. To address the reviewer's concern, we have added the above detailed simulation description in Simulations (Experimental Section, Supporting Information).

Reviewer: 2

In this manuscript Wang et al report a method of tracking the oxygen evolution reaction dynamics at the electrode-electrolyte interfaces relevant for water-splitting through a direct plasmonic sensing in Cu NPs. With the method they are able to assign and distinguish different processes that occur and later use this knowledge to rationally design a scanning strategy to improve the current density in an OER by 20%.

I found the work and concept to be intriguing which really demonstrates the superiority of a plasmonic sensing to disentangle a complex process. I also found the work to be highly multidisciplinary and will attract readers from multiple fields. I also commend the number of different analysis/experiments done to support their findings.

Having said that, unfortunately I found that the manuscript in its current state to be far from suitable for publication in Nature Communications, with reasons I will detail later below. However, I personally would be interested to see the manuscript be improved since I think it has a high potential.

Reply: We are deeply thankful to the reviewer for her/his constructive evaluation and appreciation on this work. Following these valuable suggestions, we have addressed the relevant concerns and made corresponding modifications.

Questions/recommendations:

1. Page 3: "Gas evolution leads to an undesirable blockage." What gas? What kind of evolution? How specifically does the blockage happen?

Reply: Many large-scale industrial electrochemical processes, such as aluminum production, chloro-alkaline and electrolysis of water, involve gas-evolving reactions (H_2 , Cl_2 and O_2). The heterogeneous nucleation of gas molecules at electrode surface accelerates the formation of bubble, which subsequently occupies the active sites for proceeding electrochemical reaction and blocks ion conducting pathways in the electrolyte. Therefore, the formation of bubbles results in a reduction of active electrode area, an increasing of ohmic resistance and a formation of undesirable concentration gradients. These induced effects lead to energy losses and reduction in the efficiency of electrochemical system. More detailed discussions of the

mechanism of bubble formation and corresponding influence on electrochemical processes could be found in recent review papers (*Gas bubbles in electrochemical gas evolution reactions. Langmuir, 2019, 35, 5392-5408; Electrochemistry of nanobubbles, Current Opinion in Electrochemistry, 2020, 22, 102-109; Influence of bubbles on the energy conversion efficiency of electrochemical reactors, Joule, 2020, 4, 555-579*).

2. Page 3: "at various polarization." Polarization is often mentioned but it is never explain what polarization is this. See also Page 5 "severe polarization effect".

Reply: (1) Here, the polarization refers to a shift of electrode potential from an equilibrium value. It is well known that electrochemical reactions occur in a series of steps at the electrode-electrolyte interface (*The definition of polarization, overvoltage and decomposition potential, Transactions of The Electrochemical Society, 1934, 66, 359; On the evolution and application of the concept of electrochemical polarization, Journal of Solid State Electrochemistry, 2020, 24, 2595–2602*). For instance, oxygen evolution reaction involves the diffusion of hydroxyl ion from the bulk solution close to the electrode surface, and then electrons transfer to anodic polarized electrode to form oxygen molecules. Oxygen evolution occurs when the potential of working electrode is increased to surpass the equilibrium potential of O_2/OH^- . This process could be described as an anodic polarization. When potential induced cation transportation close to the cathodic polarized electrode surface and a reduction reaction occurs at electrode surface, we can describe these processes as a cathodic polarization. In short, polarization in electrochemistry means the shift of working electrode potential from its equilibrium value to a desired value. We can describe electrode interface processes proceed under anodic polarization when the potential increases (anodic direction scan); under cathodic polarization when the potential decrease (cathodic direction scan).

(2) The severe polarization effect shown in page 5 refers to the shift of working electrode potential far from its equilibrium value when the electrochemical product accumulated at the EEs. This potential shift, namely concentration polarization, results in conspicuous suppression on oxygen evolution reaction. A more detailed discussion of the concentration polarization was shown in Supporting Information (Section 7). For more clarity, it has been

revised as “*severe concentration polarization*” (Page 4).

3. Page 4: "advanced plasmonic nanostructures" What does advanced here refer to? Any metal nanostructures whether they are advanced or not still support plasmonics.

Reply: This expression has been revised as “...*rational designed plasmonic nanostructures*...” in page 4, which means by using rational designed plasmonic nanostructures, it enables the plasmonic-enhanced spectroscopy to monitor a variety of chemical processes, including heterogeneous catalysis, electrochemistry and phase transition.

4. Page 4: "environment refractive sensitive plasmonic" The authors often use otherwise common terms in the field with their own style. This makes me confuse whether or not I am thinking what the authors are, or they are actually entirely different things. In this case I believe they meant to be "refractive-index sensitive"? Other cases include "near-plasmonic enhancement". I guess this was supposed to be "near field"? Please pay high attention to these well-known terms.

Reply: Thank the reviewer for her/his careful read and valuable suggestion. As suggested by the reviewer, we have replaced “*environment refractive sensitive*” with the more clarified term “*refractive-index sensitive*”. Meanwhile, the expression such as “*near-plasmonic enhancement*” has been modified as “*near field*”.

5. Page 7: Is CTP analogy rightly assigned to the studied system here? CTP is formed through a conductive junction between 2 particles but here instead the connection was through the metallic base and therefore not really a junction.

Reply: A conductive conjunction between two separated plasmonic nanoparticle or antenna arms could introduce a level of optical tuning control that enables precise shifts in mode frequency and the creation of optical modes (*Charge transfer plasmons: Recent theoretical and experimental developments, Applied Physics Reviews, 2017, 4, 021104*). Upon the formation of a conductive conjunction, a charge transfer plasmon (CTP) mode emerges and supports a dipolar charge configuration that distributed throughout the entire bridged

nanostructures (*Electrochemically programmable plasmonic antennas, ACS Nano 2016, 10, 6716–6724*).

The formation of a CTP, however, is regardless of connection pathways (*Charge transfer plasmons: optical frequency conductances and tunable infrared resonances, ACS Nano, 2015, 9, 6428-6435*). For example, Halas and co-workers recently reported a CTP example of dimer- on-metallic substrate, its optical properties are changed profoundly and supports a lower energy CTP mode (*Monolithic metal dimer-on-film structure: New plasmonic properties introduced by the underlying metal, Nano Letters, 2020, 20, 2087-2093*). The present system, electrodeposited copper dimers on conductive film, is quite similar to the example mentioned above, thus we assigned its plasmonic response to CTP mode with confidence.

6. Figure 1: One of the major flaws in the manuscript is how they model/simulate their particles. From the schematic and AFM (panel c and d) authors always draw their particles as a dimer with a certain gap distance. Hence all of the simulation throughout the manuscript use this dimer model. However looking at the Figure 1b this is not at all the case. Dimer was just a few percentage of the overall sample. The rest are randomly comprised of arrays of random particle, sometime even a formation of "chain-like" structures. I suggest authors to take a major change to this. Probably get a statistics on how many actually form dimer/trimer/etc, what are the average particle-to-particle distances, etc. Otherwise all of the simulation lose their credibility in my opinion.

Reply: We thank the reviewer for this thoughtful comment.

(1) Here, a seed-mediated electrochemical deposition strategy instead of electron beam evaporation or other photolithography techniques, was used to fabricate the particles-on-film structure. Due to the intrinsic limitation of non-template electrochemical method, the formation of dispersed particles and trimers is not avoidable. We agree that other structures, such as dimer and trimer on the film, could also attribute to the optical properties of the ensemble system.

We compared the distribution and amount of isolated dimer and isolated trimer (per μm^2)

on Cu NPs/ITO electrode (150 cycles). Statistical analysis indicated that the isolated dimer was more widely distributed, and its cover density (per μm^2) was *ca.* 4.5 times higher than that of isolated trimer (Figure R14, Dimer (31+28) / Trimer (7+6) \approx 4.5). Considering the higher cover density and limited configurations of isolated dimer, we chose the isolated dimer as typical individual unit for simulation. As demonstrated in the simulation results (Figure S8), we found a similar evolution trend as a function of particle size, which was in good agreement with experimental measurements (Figure 1e, main text). Thus, we tentatively concluded that the proposed model of isolate dimer-on-conductive substrate structure could represent key spectral feature of the ensemble system.

Nonetheless, we admit a more careful considering of the isolated trimer or more complicated chain structure could move the simulation results even closer and more compatible to real optical performance of the ensemble system. Considering the possible multi-configurations of the trimer (Figure R14) or chain structure, a more detailed distinguish their contribution to the ensemble system is far beyond the aim of present study, which needs more complicated modeling and combination of single particle spectroscopy techniques to correlate both simulation and experimental results.

To address the reviewer's concerns, we added a statement that "*Despite further considering the isolated trimer or more complicated chain structure could make the simulation results closer or more compatible to the real optical performance of the ensemble system, the possible multi-configurations of these structure result in an extreme difficulty in precise distinguishing their contribution to ensemble optical performance at present stage.*" in the main text (Page 7).

(2) Following the valuable suggestion from the reviewer, we provided a statistical analysis on the average particle-to-particle distances (Figure R15). According to the definition of gap, only the two adjacent NPs with a minimum distance comparable to surrounding particles are considered as a candidate dimer for statistical analysis. From the statistical analysis we found a decrease of the gap from 479 nm to 55 nm with deposition cycles from 5 to 150 cycles accomplished with a gradual growth of particle size, and a slightly increase of gap from 55 nm to 97 nm with deposition cycles from 150 to 300 cycles caused by the Ostwald ripen-like

effect (*Effect of size, coverage, and dispersity on the potential-controlled Ostwald ripening of metal nanoparticles, Langmuir 2019, 35, 16416–16426*). In order to eliminate the contribution from random dispersed particles induced over-evaluation on the gap distance, we adopted half of the gap distance (26 nm for Cu NPs/ITO, 150 cycles) in simulation. From the simulation results (Figure S8), similar spectral evolution trend was achieved compared to experiment measurements (Figure 1e).

Figure R14. (a) and (b) Representative AFM images of Cu NPs at different deposition cycles (150 cycles) (scale bar: 1 μm). The isolated dimer structure was marked by white rectangle (the amount of marked isolated dimer: a 31; b 28). (c) Magnified images of representative dimer structure (i-iii) (scale bar: 200 nm). (d) and (e) Representative AFM images of Cu NPs at different deposition cycles (150 cycles) (scale bar: 1 μm). The isolated trimer structure was marked by white rectangle (the amount of marked isolated trimer: d 7; e 6). (f) Magnified images of representative trimer structure (i-iii) (scale bar: 200 nm). (g) The spatial configuration of both dimer and trimer. Owing to the increase of the symmetry of trimer, more possible configurations are present in the ensemble system, which result in the extreme difficulty in distinguish their contribution to optical characteristics.

Figure R15. (a) Representative AFM images of Cu NPs at different deposition cycles (i 5, ii 50, iii 150 and iv 300 cycles) (scale bar: 1 μm). (b) The corresponding gap distribution (peak location and full width at half maxima, FWHM) of electrodeposited Cu NPs. The minimum gap between two adjacent nanoparticles was chosen for statistical analysis.

7. Figure 1f: which one is top and which one is side? They need to be clearly labeled. Also in both cases they look round? Shouldn't the one from the side look hemispherical?

Reply: Figure 1f only represents the *top-view* of electric and magnetic field snapshots for the near field confinement at the plasmonic resonance wavelengths. The corresponding modification has been performed in caption of Figure 1f as “(f) *Top-view of electric (i and ii) and magnetic (iii and iv) field snapshots for the near field confinement at the plasmonic resonance wavelengths (i, iii 450 nm and ii, iv 690 nm).*”.

8. Page 8: "we observed a remarkable increase..." and the rest of the paragraph. I can't really see what the authors described here in the data. The way Figure 2b is plotted does not help as well.

Reply: Figure 2b shows the color-code spectral evolution of Cu NPs/ITO electrode under electrochemical steady-state condition. To make it clear, we displayed the corresponding potential-dependent spectra in an anodic potential scan from -1.10 to $+1.10$ V in Figure R16. The modified Figure was supplementary in Supporting Information (Figure S11).

Here we aimed to use electrochemical steady-state extinction spectroscopy to monitor

OER at electrode surface (Figure 2b). With the enhancement of anodic polarization, the spectral intensity of Cu NPs/ITO was decreased gradually. The dynamic spectral changes indicated that the Cu oxide and hydroxides intermediates were weakly bound and readily dissolved during long time anodic polarization, which hampered interfacial processes dissection. Thereby, it prompts us to develop an alternative single-wavelength plasmonic monitoring strategy to precisely identify the diffusion behavior of dissolved oxygen to avoid the persistent oxidation of the electrode surface under electrochemical steady-state condition.

Figure R16. (a) The normalized dynamic extinction intensity (contour plots) of a Cu NPs/ITO electrode under steady-state anodic polarization from -1.20 V to $+1.10$ V. (b) The representative normalized extinction spectra at various polarized potential.

9. Page 8: "These indicate that the Cu oxide and..." What is the base for this statement? There is no reference and there is data/simulation by the authors to support this.

Reply: We are sorry for missing references on this statement. Relevant references (*Ellipsometric Study of oxide formation on Cu electrode in 0.1 M NaOH. J. Solid State Electrochem. 2009, 13, 1639-1644; Cyclic voltammetry of copper in sodium hydroxide solutions. J. Electroanal. Chem. Interfacial Electrochem. 1981, 117, 309-319*) were added. Moreover, the gradual attenuation of spectral intensity following long time of anodic potential scan also indicated the dissolving of the Cu oxide and hydroxides intermediates under electrochemical steady-state condition (Figure R16).

10. Figure 2c is unclear and hard to understand. What are the colors?

Reply: We thank the reviewer for her/his careful read and thoughtful comments. The scale

bar of the color has been added in Figure 2c. Here the color represents the spectral intensity at various wavelength. The electrochemical reactions at the electrode-electrolyte interface could lead to the fluctuation of the spectral intensity, which enables us to in situ track the intensity at single wavelength to achieve the interfacial reaction information.

11. On using the single wavelength extinction spectroscopy, I did not find anywhere about details regarding the experiment. What wavelength did the authors used? What was the reason on using that wavelength? Have comparison to other wavelengths been conducted?

Reply: We thank the reviewer for this thoughtful comment.

(1) The detailed experimental descriptions on the optical apparatus applied for acquiring dynamic extinction signals were added in the Instrument Section (main text) as:

“To in situ record the spectral signals of Cu NPs/ITO electrode during dynamic electrochemical reaction, the halogen lamp and deuterium lamp were served as the light source. We used Czerny-Turner type monochromator (lens 1300 lines mm⁻¹) to provide monochromatic light. A photomultiplier tube (PMT) was used as detector to collected the optical signals from the EEIs.” (Page 20-21).

(2) Electrochemical seed-mediated growth of Cu NPs was adopted to fabricate plasmonic-enhanced system for single-wavelength monitoring. With the enlargement of NP size, the extinction cross section was increased gradually and reached saturation after deposition 150 cycles (Figure 1, shown in main text). Moreover, the as-prepared Cu nanostructures (deposition 150 cycles) exhibit a pronounced CTP mode at 690 nm (Figure 1e). Therefore, the electrodes obtained by electrochemical deposition for 150 cycles were used for both electrochemical and optical measurements. Here, we monitored the temporal extinction trajectory of Cu NPs/ITO electrode at 690 nm during dynamic potential scan. To make it more explicit, we have added the above statements in the Instrument Section (main text, Page 20-21).

(3) According to the reviewer’s suggestions, we compared the different temporal extinction trajectories at various wavelength during consecutive potential scan (Figure R17, this figure was also added in the Supplementary Information as Figure S12). The electrodeposited Cu

NPs electrode exhibited no obvious spectral fluctuation at open circuit potential in 90 min (Figure R17 a), suggesting excellent stability. Different wavelengths (400, 443, 560, 620, 690, 770 and 850 nm) were chosen for comparison the sensing performance of OER (Figure R17 b). When wavelength located at the CTP mode (690 nm) was adopted for monitoring, its spectral response exhibited distinct enhancement in intensity compared with others (Figure R17 c-e). For instance, the intensity increased *ca.* 1.60 and 2.33 times at 690 nm (CTP) compared to that of 443 nm and 560 nm, respectively. Therefore, tracking the electrochemical interfacial reaction at 690 nm (CTP) enables explicitly identifying characteristic spectral regions for revealing the OER mechanism. The sensitive interfacial reaction sensing response at the CTP mode is attributed to its intrinsic vulnerability to dynamic environment refractive index (*Monolithic metal dimer-on-film structure: New plasmonic properties introduced by the underlying metal, Nano Letters, 2020, 20, 2087-2093*).

Moreover, control experiments on bare ITO electrode monitored at 400 nm and 443 nm, revealed weak periodic fluctuation of background signal, attributing to mass transfer of interfacial species (K^+ , OH^-), while no obvious fluctuation of background signal was observed at 690 nm. The importance of CTP mode is clearly highlighted by both high sensitivities to interfacial reaction and negligible background signals.

Figure R17. (a) The stability of the extinction spectrum of a representative Cu NPs/ITO electrode at open circuit potential during 90 min. (b) The locations of different wavelength (I 400, II 443, III 560, IV 620, V 690, VI 770 and VII 850 nm) adopted for single wavelength monitoring. (c) The comparison of the maximum spectral intensities at various monitoring wavelength (I-VII, 400 nm-850 nm). The consecutive electrochemical current (i) of a Cu NPs/ITO electrode (d) and a bare ITO electrode (e) during OER and synchronous extinction signal (ii-viii, 400 nm-850 nm) under dynamic potential scan (ix) from -1.10 to 0.90 V at a scan rate 0.01 V s^{-1} .

12. Page 10: On the five typical potential regions (Figure 2e), it is unclear whether these 5 regions are well-known from the literature or it is derived from the authors observation? Because if it is from the former, I don't see any references. This is important because later in the paragraph they assign the change of the optical signal to different mechanisms that occur based on these 5 regions.

Reply: Thank the reviewer for careful read and thoughtful comments on our manuscript. Just as the statement of the reviewer, it is important to distinguish these 5 characteristic regions. These five regions were derived from our observation from both electrochemical and spectral measurements. Based on the temporal extinction trajectory and synchronized current response, five typical potential regions were identified to explain the dynamic interfacial processes (Figure 2e).

These five regions were: (I) active region, where the oxidation of Cu occurs; (II) passive region, where oxygen evolution is suppressed; (III) transpassive region, where the onset oxygen evolution region is controlled by reactive kinetics; (IV) diffusion region, where oxygen diffusion regions is dominated by diffusion kinetics and (V) reductive region, where oxygen reduction occurs.

(1) The detailed discussion on identification of region I, II and III was shown in Supplementary Section 4 and Figure S9 (Page S18-S20). To take the reviewer comment into account, the corresponding discussion and Figure S9 was shown below:

Figure S9. Oxygen evolution reaction on Cu NPs electrode. (a) The linear scan curves for a Cu NPs/ITO electrode in 0.10 M KOH solution with various termination potential (a-h: 0.50, 0.60, 0.70, 0.80, 0.90, 1.00, 1.10 and 1.20 V), at a scan rate of 0.01 V s⁻¹ (the graphs were vertically shifted for ease of comparison). Inset: the band gap transition in the transpassive region. The schematic illustrations of various interfacial processes during oxygen evolution reaction (i-iii). (b) Scan rate dependence of CV curves for a Cu NPs/ITO electrode in a solution of 0.10 M KOH. Scan rate: 0.01, 0.02, 0.03, 0.04, 0.05, 0.06, 0.07, 0.08, 0.09, 0.10, 0.20, 0.30, 0.40, and 0.50 V s⁻¹. (c) The dependence of anodic peak current at peak A1, A2, and A3 on scan rate. (d) The dependence of cathodic peak current at peak C1 and C2 on the square root of the scan rate.

“Linear potential scan measurements were used to track dynamic interfacial process at the EEs (Figure S9). The dynamic interfacial process exhibited three characteristic

regions with increasing potential: (1) At the interface active region (< 0.30 V vs. Pt unless otherwise stated, all the potential referenced to a Pt quasi-reference electrode), the process of OH⁻ adsorption took place, assigning to the peak located at $- 0.63$ V (peak A1). With the increasing of potential to greater than $- 0.39$ V, the adsorbed phase on the copper surface eventually reconstructed, resulting in the formation of a Cu₂O layer, which was indicated by the presence of peak A2. (*J. Solid State Electrochem.* 2009, 13, 1639-1644) With further enhancement in anodic polarization, the arising of broad peak A3 was attributed to the accompaniment of both electrooxidation of Cu(I) to Cu(II) and the formation of CuO and Cu(OH)₂ (Region i, a detailed kinetic analysis shown in Figure S9); (2) At the interfacial passive region (0.30 V \sim 0.60 V), the formation of a thicker and complex Cu oxides and hydroxides passive film restrained the interfacial oxidation, and no obvious catalytic activity towards oxygen evolution was observed (Region ii); (3) At the interfacial transpassive region (> 0.60 V), the gradually enhanced dehydration process accompanied by a structural transformation from an amorphous to a crystalline structure, providing an electron-conductive pathway for oxygen evolution (Region iii). The crossing of the CuO/Cu(OH)₂ layer valence band and the Fermi level caused an enrichment of positive holes in this transpassive region, that significantly improved the oxygen evolution efficiency (Figure S9, a inset). The proposed interfacial reaction mechanisms on Cu NPs electrode in alkaline solution were expressed as:

The influence of the sweep rate on both anodic and cathodic polarization was shown in Figure S9, b. The cathodic peaks C1 was assigned to the electrochemical reduction of CuO to Cu₂O. The other peak C2 was caused by the simultaneous reduction of both Cu₂O and CuO to Cu. (*J. Solid State Electrochem.* 2009, 13, 1639-1644) During kinetic analysis, a linear relationship between anodic peak currents and the scan rate indicated a

surface reaction controlled kinetic in anodic processes (Figure S9, b, Peak A1, A2 and A3). Meanwhile, the cathodic processes (Peak C1 and C2) exhibited a linear function as the square root of the scan rates, indicating a diffusion-controlled mass transport in these processes (Figure S9, c). Interestingly, a gradually emerging cathodic peak located at -0.45 V was found at relative high scan rate ($> 0.20\text{ V s}^{-1}$) which was assigned to the reduction from $\text{Cu}(\text{OH})_2$ to Cu_2O . Taken together with kinetic analysis of interfacial reactive species, the linear scan voltammetry enabled to reasonably elucidate their compositions and distinguish controlled manner of interfacial reactions under polarization.”

- (2) The temporal extinction trajectory (Figure 2e) enables us to identify the region IV and V, which was unavailable from traditional electrochemical measurements. Both IV and V regions that directly derived from in situ extinction measurements were assigned to the oxygen diffusion at the EEIs. It should be noted that the present rational designed plasmonic nanostructure enables us to monitor the dynamic diffusion behavior of oxygen at the electrode surface without obvious interference from the non-Faradaic processes that unavoidable in electrochemical measurements.

13. Related to point above, the authors then rationalized the increase/decrease in the extinction to process occurring at the interface. One example, in Region 1 the phase growth within an oxide layer is ascribed to the increase of extinction intensity. For all of these I think the authors should try to verify this through their FDTD. Also, again, depending on which wavelength the extinction is taken with respect to the position of the LSPR peak, the response can take form in the increase or decrease. As such all of these correlation between the extinction intensity and processes occurring at the interface are premature.

Reply: We thank the reviewer for her/his careful read and thoughtful comments on previous manuscript. We performed more control experiments and simulation to make the division of different characteristic region more concrete.

- (1) The Cu_2O growth is accomplished by a phase transition between copper oxide and hydroxide occurred at the electrode surface, which will cause variations of thickness. Thus,

we also simulated the optical response of dimers with different thickness of Cu_2O layer (the thickness of Cu_2O layer was rationally set from 1 nm to 4 nm in potential range from -1.10 V to -0.36 V; then decrease back to 1 nm in potential range from -0.36 V to -0.10 V). As shown in Figure 2e (Region I corresponding to peak a), we found the evolution of simulated extinction intensity similarly to experimental results.

In this region, the composition of electrode interface suffers great changes during dynamic potential scan, including interfacial composition dynamics, phase transition and surrounding mass transfer. Because of these complexities, we considered only the contribution from Cu_2O layer and ignored both copper oxide and hydroxide to extinction intensity. We confess that a more complicated model with higher temporal and chemical resolution may help to deeply reveal the interfacial reaction, but it far beyond current capability.

(2) As suggested by the reviewer, we have experimentally tracked the evolution of extinction intensity at various wavelength (Figure R17). From these spectral evidences, the sensing at 690 nm (CTP) exhibits a higher performance. We thus concluded that tracking the electrochemical interfacial reaction at 690 nm (CTP) enables explicitly identify the characteristic spectral regions for revealing the mechanism of the oxygen evolution reaction.

(3) Based on these findings, we proposed a programmable potential scan strategy to eliminate the concentration over-potential induced by accumulated oxygen at the electrode-electrolyte interface (Figure 5, main text). All these experimental results provide more direct evidences to corroborate the reasonability of our proposed multi-diffusion behavior of oxygen during OER.

14. Page 12: Where do the refractive index of 2.9, 2.72 and 1.71 for different Cu state come from? Literature of authors' own experiments?

Reply: The corresponding references have been added in the main text as “*As the enhancement of anodic polarization, the phase transition from Cu_2O to CuO and $\text{Cu}(\text{OH})_2$ phases led to the reduction of the refractive index from 2.90 to 2.72 and 1.71, resulting an increasing of the extinction intensity. (Ribbing, C. G.; Roos, A., Copper Oxides (Cu_2O , CuO).*

Handbook of Optical Constants of Solids, Palik, E. D., Ed. Academic Press: Boston, 1998; 875-882.; Batsanov, S. S., Ruchkin, E. D., & Poroshina, I. A., *Refractive Indices of Solids*, Springer: Singapore, 2016; 85-100.)". (Page 11)

15. Page 12: Authors argue that the dissolved oxygen accumulated at the electrode to cause reduction of local RI, based on their simulation shown in Fig. S21. The problem here is that the simulation condition is unclear. Instead of technical schematic of how the structures were simulated, there is only an artistic figure of the system. How did the authors simulate the gradient of the O₂ concentration? What is the input for their refractive index? This part should be significantly amended.

Reply: During OER, oxygen molecules evolved through electrolysis of water ($4\text{OH}^- \rightarrow 2\text{H}_2\text{O} + \text{O}_2$) and gradually accumulated at the EEs. As the reviewer's statement, it's true that the evolved oxygen will diffuse to bulk electrolyte solution and form a diffusion layer around electrode.

We have also demonstrated the simulated oxygen concentration profiles along the normal direction toward to the bulk electrolyte solution at various potential scan rate in Figure 4f (main text) according to Butler-Volmer equation and Fick's 2nd law (*Pore confined liquid-vacuum interface for charge transfer study in an electrochemical process, Anal. Chem. 2019, 91, 3195-3198*). From these numerical simulations, we could see a remarkable nonlinear distribution along the oxygen concentration profiles. Therefore, it is very hard to make an accurate description of dynamic oxygen concentration profiles with FDTD simulation. Instead and as a compromise, we adopted two strategies to approximately estimate the influence of oxygen accumulation on the optical properties based on the individual unit cell dimer-on-film structure shown in Figure S8.

(1) As oxygen evolution, the refractive index of the electrolyte (1.33) located at interfacial region could gradually decrease due to the accumulation of oxygen molecules (refractive index: 1.00, *Precision refractive index measurements of air, N₂, O₂, Ar, and CO₂ with a frequency comb. Applied optics, 2008, 47, 3143-3151*). Because the limitation in precise identification of absolute oxygen concentration close to the electrode surface, it is very hard and

impossible to acquire the dynamic hybrid *refractive index* of interfacial electrolyte (including solvated oxygen) during dynamic potential scan. To qualitatively evaluate the influence of oxygen accumulation, we rationally set the refractive index of environment from 1.33 to 1.06 in simulating the dynamic environment in potential range from + 0.20 V to + 0.90 V then to + 0.67 V (Region III corresponding to 130 s to 220 s in time scale); To further evaluate subsequent oxygen spontaneous and accelerated diffusion processes, the refractive index of environment was rationally set from 1.06 to 1.20 in potential range from + 0.67 V to – 0.35 V (Region IV corresponding to 220 s to 319 s in time scale) and from 1.20 to 1.33 in potential range from – 0.35 V to – 0.63 V (Region V corresponding to 319 s to 365 s in time scale), respectively (Figure S26 a, b).

(2) We then considered an extreme condition, where oxygen molecules are accumulated on the electrode surface to form an oxygen shell around Cu NPs, this enables to exclude surrounding aqueous electrolyte solution and to decrease the refractive index of environment. To qualitatively evaluate the influence of oxygen accumulation, we rationally set the thickness of the oxygen shell from 0 nm to 120 nm to simulate the dynamic environment in the potential range from + 0.20 V to + 0.90 V then to + 0.67 V (Region III corresponding to 130 s to 220 s in time scale); To further evaluate subsequent oxygen spontaneous and accelerated diffusion processes, the thickness of oxygen layer was rationally set from 120 nm to 85 nm in the potential range from + 0.67 V to – 0.35 V (Region IV corresponding to 220 s to 319 s in time scale) and from 85 nm to 0 nm in potential range from – 0.35 V to – 0.63 V (Region V corresponding to 319 s to 365 s in time scale), respectively (Figure S26 c, d).

Despite all these efforts, it should be noted that the dynamic of oxygen diffusion at the EEIs is far more complicated than what we proposed here. Our simulation results only provide a qualitative evaluation of interfacial processes, a more precise description of the dynamic interfacial electrochemical reactions should consider electric field interference, build more complicated models and adopt density functional methods or molecular dynamics simulations.

In order to address the reviewer's concerns, we have made corresponding modifications and removed the misleading word "*gradient*" from the caption in Figure S21 (It has been re

numbered as Figure S26). Moreover, we have added the above detailed simulation description in Simulations (Experimental Section, Supporting Information, Page S48).

16. Page 13: "obsolete potential-dependent relationship". What is the context of the obsolete here?

Reply: Here, we used different terminational potential in anodic potential scan and found that the extinction peak b increased quickly with enhancement of anodic polarization (> 0.50 V, Figure 2g). It indicates an evident accumulation of dissolved oxygen at the EEs under OER condition. Meanwhile, the extinction peak a that assigned to the electrogenerated Cu oxides and hydroxides showed a distinct potential-dependent relationship in which involves the transformation from Cu_2O to CuO and $\text{Cu}(\text{OH})_2$, and subsequent deprotonation of the $\text{Cu}(\text{OH})_2$. For more clarity, it has been revised as “...*distinct potential-dependent relationship...*” (Page 12).

17. Page 14: "rather than other gas, for example, air". Air is composed of 20% O_2 , so in this context air and oxygen are the same. Authors should trace with their MS other relevant gases that occur in the reaction, not just H_2O like the show in Fig. 3.

Reply: In order to confirm the gas product during electrochemical interfacial reaction, we used differential electrochemical mass spectrometry (DEMS) to provide direct molecular evidence for gas product. From DEMS results we could see a clear increase of oxygen intensity during dynamic potential scan, therefore oxygen is confirmed as the electrochemical catalytic product. Based on the law of conservation of mass and elements, no other elements species were present in 0.1 M KOH aqueous electrolyte solution except H^+ , OH^- and K^+ , and therefore other gases cannot be generated under anodic conditions. To make it more clarity, this sentence has been revised as “*We further sought mass spectrometer to provide direct molecular evidence to solidify that dynamic extinction spectra variations were results of electrochemical-generated oxygen.*” (Page 12).

18. Figure 5a: Please add legends for all the symbols.

Reply: The legends for all the symbols have been added in Figure 5a for more clarity (Page 16).

19. Lastly, the manuscript is unfortunately not great in terms of its language. It is riddled with grammatical errors and inappropriate terms that make it very difficult to grasp the idea efficiently. One prime example, even the title is not grammatically correct. I strongly recommend the authors to take a significant (professional) proofreading for all of the texts (main text and SI).

Thank you very much and good luck.

Reply: We thank the reviewer for her/his careful read and thoughtful comments on our revised manuscript. According to the reviewer's suggestion, a professional English editor has read the manuscript. The language has been polished and many grammatical issues were corrected. The more detailed modifications were marked in red color in the manuscript.

Reviewer: 3

In this manuscript, the authors proposed an in situ spectroelectrochemical method to uncover the multi-stage diffusion of oxygen molecules during OER catalyzed by copper nanoparticles. Guided by the inhibition effect of oxygen accumulation they discovered, they further improved the OER performance through a rationally-designed potential scan sequence. Their spectroelectrochemical method relied on the sensitive dependence of the plasmonic spectrum of a semi-transparent ITO substrate (which was pre-electrodeposited with copper nanoparticles) on the refractive index of electroactive surface (copper nanoparticles here) as well as the surrounding medium (oxygen molecules in this case).

The general impression from this reviewer is that, the study was nicely designed and conducted. In addition to the quite impressive data quality and in-depth discussion, this reviewer particularly likes the last part on the improved OER performance. While many studies in the field stopped at the mechanism clarification, this work sets a rare example showing how mechanism advancements indeed helped the performance improvements. This reviewer thus supports the publication of this study in Nature Commun if the authors can address the following points.

Reply: We are deeply thankful to the reviewer for her/his constructive evaluation and appreciation of this work. Following your valuable suggestions, we have made corresponding modifications in our manuscript to address the reviewer's concerns.

1. While this study relied on the extinction spectrum (and intensity at single wavelength), experimental descriptions on the optical/spectral apparatus are almost completely missing (Fig. 1a). The light source (LED? Lamp? White Laser?), the spectrometer, the detector (DAD? PD?) must be specified. Whether or not an objective/lens was used in the light path to collect the light?

Reply: As suggested by the reviewer, the detailed experimental descriptions on the optical apparatus applied for acquiring dynamic extinction signals have been added in the Instruments Section (main text) as:

“To in situ record the spectral signals of Cu NPs/ITO electrode during dynamic

electrochemical reaction, the halogen lamp and deuterium lamp were served as light source. We used Czerny-Turner type monochromator (lens 1300 lines mm⁻¹) to provide monochromatic light. A photomultiplier tube (PMT) was used as detector to collected the optical signals from the electrode-electrolyte interface.” (Page 20).

2. In Fig. S7c, HR-TEM image was given. It was unclear how the authors prepared the sample (transfer from ITO substrate to TEM grid)? A ~1 nm-thick shell was identified from the image. Please explain.

Reply: (1) The Cu NPs were electrodeposited on the ITO substrate with strong adhesion. Therefore, it cannot be collected and imaged for TEM measurement as traditional colloidal solution. The detailed experimental description on how to prepared the sample for TEM characterization has been given in the main text (Instruments Section) as:

“For TEM characterization, the electrochemical deposited Cu nanoparticles were gently scraped by a steel blade from the ITO electrode and subsequently dispersed in anhydrous acetonitrile solution by sonication (20 s). Then, a droplet of the suspension with 20 μ L was gently dropped onto a carbon coated-copper grid (300 mesh), allowing the solvent to evaporate prior to imaging.” (Page 20).

(2) The shell around the copper nanoparticle was slight surface oxides which caused by the exposure to air during transport to the TEM. Compared the lattices of shell region (R1) and central section (R2) of Cu nanoparticle, Figure R18 exhibits lattice spacing of the shell region corresponding to Cu₂O(111) spacing (0.24 nm) and Cu(111) spacing (0.21 nm), respectively (*Visualizing the Cu/Cu₂O interface transition in nanoparticles with environmental scanning transmission electron microscopy, J. Am. Chem. Soc. 2017, 139, 1, 179–185*). The two spacing of shell and central regions are directly related to each other with parallel epitaxy between the Cu and Cu₂O *fcc* crystal structure. A similar phenomenon for the air exposure induced oxides formation has been recently reported in both Christoph’s (*Copper catalysis at operando conditions—bridging the gap between single nanoparticle probing and catalyst-bed-averaging, Nat. Commun. 2020, 11, 4832*) and Beatriz’ work (*Improved CO₂ electroreduction performance on plasma-activated Cu catalysts via electrolyte design: halide*

effect, *ACS Catal.* 2017, 7, 8, 5112–5120).

To probe the interface of the Cu nanoparticle, high-sensitivity energy-dispersive X-ray spectroscopy (EDS) and scanning transmission electron microscopy (STEM) were introduced to further probe the interface of the Cu NPs. Figure R19 shows a relative weak oxygen element distribution around Cu NPs, while obvious oxide layer around the Cu NPs was not observed. Moreover, the lattice spacing of 0.21 nm was corresponded to the (111) spacing of Cu. Even so, we cannot completely exclude the possibility of fast oxidation of metallic Cu during the transfer in air to the TEM chamber. (*Oxidation of polycrystalline copper thin films at ambient conditions*, *J. Phys. Chem. C* 2008, 112, 4, 1101–1108). In order to make it more clarification for readership, we added the supplementary sentence “*The slight surface oxides was caused by exposure to air during transport to the TEM.*” in the caption of Figure S7 (Page S16).

Figure R18. (a) The enlarged high resolution TEM image shown in Figure S7c and the magnified shell region (b) and central region (c) in copper nanoparticle. (d) Line intensity profiles from the green and red boxes in (b) and (c) with annotations indicating the lattice spacing 0.24 nm corresponding to $\text{Cu}_2\text{O}(111)$ and 0.21 nm of $\text{Cu}(111)$, respectively. The slight surface oxides were caused by exposure to air during transport to the TEM

Figure R19. Annular bright-field (ABF) STEM (a), medium-angle annular dark-field (MAADF) STEM (b), and high-angle annular dark-field (HAADF) STEM image (c) of Cu NPs (after electrodeposited 150 cycles). (d-f) The EDS element mapping images of Cu NPs corresponding to image (c). The oxygen element shows a random distribution pattern around Cu NP. It can be tentatively attributed to the contribution from air exposure during sample transport. (g) High resolution image of Cu NP and the enlarged square region was shown in (h). (i) Line intensity profiles from the green and red boxes in (h) with annotations indicating the lattice spacing 0.21 nm corresponding to Cu(111), respectively.

3. In Fig. 1b, it is recommended to display potential-dependent spectra in a complete cycle from -1 to 1 to -1 again.

Reply: The reviewer might refer to the Fig. 2b in which we showed the color-code spectral evolution of Cu/ITO electrode under electrochemical steady-state condition. As suggested by the reviewer, we have displayed the corresponding potential-dependent spectra in an anodic potential scan from -1.10 to $+1.10$ V in Figure R20 (This figure was also added in the Supplementary Information as Figure S11). With the enhancement of anodic polarization, the intensity of Cu NPs/ITO was decreased gradually and the dynamic spectra indicated that the

Cu oxide and hydroxides intermediates were weakly bound and readily dissolved during long time anodic polarization, which hampered interfacial processes dissection under steady-state conditions. With a long time of anodic potential scan, it induced the dissolved of the Cu oxide and hydroxides intermediates, resulting in severe difficulty in acquiring reversible potential-dependent spectral behavior during subsequent cathodic potential scan. Because of this, we exhibited dynamic spectra only under the anodic potential scan to indicate the limitation when applied electrochemical steady-state measurements. Meanwhile, it prompts us to develop a single-wavelength plasmonic monitoring strategy to precisely identify the diffusion behavior of dissolved oxygen to avoid the persistent oxidation of the electrode surface (Figure 2c).

Figure R20. (a) The normalized dynamic extinction intensity (contour plots) of a Cu NPs/ITO electrode under steady-state anodic polarization from -1.20 V to $+1.10$ V. (b) The representative normalized extinction spectra at various polarized potential.

4. Is it possible for the authors to discuss on the sensing distance of Cu nanostructures to report the oxygen amount/concentration? It is clear that only the oxygen molecules nearby the nanostructure surface could contribute the extinction. But how far?

Reply: This is an interesting suggestion. In principle, oxygen was electrochemically evolved at the EEIs rather than diffusing from the bulk solution closed to the electrode surface. The dynamic of oxygen concentration at the EEIs caused corresponding spectral fluctuations. Considering the intrinsic dynamic characteristics of the oxygen evolution reaction, it is therefore an extreme challenge and very hard to in situ probe the oxygen concentration around electrode surface without appropriate internal reference used to calibrate the

extinction signals.

The finite element simulation provides different oxygen concentration profiles ranging from the electrode surface to the bulk solution at various potential scan rate (Figure R21 a). We tried to qualitative evaluation the sensing distance of Cu nanostructures to the evolved oxygen by comparison the experimental extinction results and the numerical simulations. By careful comparing the simulated oxygen concentration at different locations away from the electrode surface with the extinction intensity under various potential scan rate, it enables us to correlate the extinction intensity with the sensing distance. Figure R21 b shows the experimental extinction intensity as a function of root of scan rate. The extinction intensity follows a similar trend with the oxygen concentration located at 100 μm or less than 100 μm away from the electrode surface. Therefore, we roughly estimated a sensing distance of *ca.* 100 μm for evaluation the oxygen evolution behavior.

Figure R21. (a) Numerical simulation of concentration profiles at various polarization potential (0.25 V, 0.33 V, 0.55 V and 0.67 V) corresponding to the peak potential (peak b shown in Figure 2e) at various scan rates 0.10, 0.05, 0.025 and 0.01 V s^{-1} , respectively. (b) The comparison between the dependence of oxygen concentration at different locations away from the EEIs and extinction intensity (peak b) on the square root of the scan rate. The fitting lines show the trend of the oxygen concentration as a function of the root of potential scan rate (simulation results: solid line; extinction intensity: dash line).

5. In Fig. 2e, it remains difficult for this reviewer to understand why the onset potential in extinction signal was so much earlier than that in current signal. The authors explained that current signal was consisted of both Faradaic and non-Faradaic components. It is true. But given the quite small non-Faradaic charging background and the significant extinction signal

in the early stage in region III, there could be some other reasons behind the mismatch. This point is more concerning when looking at the MS-data in Fig 3b, red curve. The onset potential in mass signal is not consistent with that in the extinction signal, while they should be consistent because they both indicate the amount of product (oxygen molecules).

Reply: (1) We attributed the improvement of onset potential to the efficient elimination of non-Faradaic component during optical measurement. Considering the ensemble system used in our study instead of single nanoparticle or nanoelectrode system (in which no obvious non-Faradaic contribution was recorded), the obvious non-Faradaic component in electrochemical dynamic potential scan resulted in extreme difficulty to accurately distinguish the onset potential using traditional electrochemical measurements. If we apply single entity system where the electrochemical measurement can avoid the influence from non-Faradaic contribution (*Imaging the chemical activity of single nanoparticles with optical microscopy, Chem. Soc. Rev., 2018,47, 2485-2508; Plasmonic sensing and control of single-nanoparticle electrochemistry, Chem, 2018, 4, 1560-1585*), the electrochemical measurement will be equivalent to optical method in acquiring onset potential of oxygen evolution. This discussion is out of our main point for ensemble system in identification the dynamic diffusion behavior of oxygen at EEs. However, this is a good suggestion, and we can compare the onset potential in single nanoparticles system in the future.

(2) In order to confirm the gas product during electrochemical interfacial reaction, we used differential electrochemical mass spectrometry (DEMS) to capture gas product to provide direct molecular evidence. From DEMS results we observed a clear increase of the oxygen intensity during dynamic potential scan; therefore, oxygen was confirmed as the electrochemical catalytic product.

However, the onset potential between the DEMS and the extinction signal could not be synchronized. This is attributed to sampling delay effect in the DEMS measurement. The gas product at the electrode surface is sampled by the injection into mass spectrometer and transported to the mass detector. The induced sampling time results in a delay of real time probing interfacial product during dynamic potential scan, while the extinction signal does not require the sampling and transport process. Therefore, the extinction signal provides more

accurate dynamic reaction information.

Reviewer #1 (Remarks to the Author):

The authors carefully addressed all issues that the reviewer raised, and carried out extensive experiments to rationalize their results. As a reviewer, it was difficult to understand the descriptions at first. For example, the authors always wrote "Cu NPs electrode", but they were not actually Cu(0), but Cu(I) or Cu(II) under the reaction environments. The authors were clever to assign the bare surface as Cu₂O in Region I, and thus all explanations matched with the extinction intensity in Figure 2e. Can the authors clearly describe it to show that the active surface is actually copper oxide? Otherwise, the answers and additional experiments were satisfactory, and this manuscript is now suitable for publication in this journal.

Reviewer #2 (Remarks to the Author):

The authors have done an exceptional job in answering all of the reviewers' suggestions. Now the manuscript is in much better shape and certainly is worthy of publication in Nature Communication.

Congratulations for the authors, this is an exceptional work!

Nonetheless I still have a few suggestions that probably might be considered:

Point 1: Please add at least one of the invoked references into the manuscript.

Point 3: I think "rationally-designed" is more appropriate here.

Point 6: Why aren't Fig. R14 and R15 included in the revised SI? I think this information is crucial.

Reviewer #3 (Remarks to the Author):

This reviewer is satisfied with the revision and thus recommends the publication of this study in Nature Communications.

Reviewer: 1

The authors carefully addressed all issues that the reviewer raised, and carried out extensive experiments to rationalize their results. As a reviewer, it was difficult to understand the descriptions at first. For example, the authors always wrote "Cu NPs electrode", but they were not actually Cu(0), but Cu(I) or Cu(II) under the reaction environments. The authors were clever to assign the bare surface as Cu₂O in Region I, and thus all explanations matched with the extinction intensity in Figure 2e. Can the authors clearly describe it to show that the active surface is actually copper oxide? Otherwise, the answers and additional experiments were satisfactory, and this manuscript is now suitable for publication in this journal.

Reply: We are deeply thankful to the reviewer for her/his constructive evaluation and recommendation our work for publication.

During oxygen evolution reaction, copper oxides and hydroxides are gradually formed on the surface of Cu NPs and exhibit dynamic potential-dependent evolution behavior of their composition during anodic polarization (Supplementary Figure 11, 23-25). We agree that both copper oxides and hydroxides serve as real reactive species to react with OH⁻, and involve in the generation of oxygen (Figure 2, main text). According to this valuable suggestion, we have made corresponding modifications in our manuscript to address the reviewer's concern. The sentence "*Note that both copper oxides and hydroxides serve as real reactive species to react with OH⁻ and then generate oxygen.*" was added in the caption of Figure 2, in order to clarify the real reactive species, which contribute to oxygen evolution.

Reviewer: 2

The authors have done an exceptional job in answering all of the reviewers' suggestions. Now the manuscript is in much better shape and certainly is worthy of publication in Nature Communication.

Congratulations for the authors, this is an exceptional work!

Reply: We are deeply thankful to the reviewer for her/his constructive evaluation and recommendation our work for publication. Following these valuable suggestions, we have addressed the relevant concerns and made corresponding modifications.

1. Please add at least one of the invoked references into the manuscript.

Reply: According to the valuable suggestion from the reviewer, the invoked references (*Electrochemistry of nanobubbles, Current Opinion in Electrochemistry, 2020, 22, 102-109; Influence of bubbles on the energy conversion efficiency of electrochemical reactors, Joule, 2020, 4, 555-579*) were added in the manuscript (Reference 7 and 8, Page 3).

2. I think "rationally-designed" is more appropriate here.

Reply: Thank the reviewer for her/his careful read and valuable suggestion. As suggested by the reviewer, the corresponding expression was revised as "...*rationally-designed plasmonic nanostructures...*" in page 4.

3. Why arent Fig. R14 and R15 included in the revised SI? I think this information is crucial.

Reply: Fig. R14 and R15 reveals the possible multi-configurations of trimer or chain structure and provides a quantitatively description of the gap distribution of electrodeposited Cu nanoparticles, respectively. As suggested by the reviewer, these information were included in the revised Supporting Information (Supplementary Figure 9 and 10).

Reviewer: 3

This reviewer is satisfied with the revision and thus recommends the publication of this study in Nature Communications.

Reply: We are deeply thankful to the reviewer for her/his time and effort and recommendation our work for publication.